# A doxycycline- and light-inducible Cre recombinase mouse model for optogenetic genome editing

Miguel Vizoso ●[1] ✉, Colin E. J. Pritchard ●[2], Lorenzo Bombardelli[3], Bram van den Broek ●[4,5], Paul Krimpenfort[2], Roderick L. Beijersbergen ●[3,6], Kees Jalink ●[4,7] & Jacco van Rheenen ●[1] ✉

The experimental need to engineer the genome both in time and space, has led to the development of several photoactivatable Cre recombinase systems. However, the combination of inefficient and non-intentional background recombination has prevented thus far the wide application of these systems in biological and biomedical research. Here, we engineer an optimized photo-activatable Cre recombinase system that we refer to as doxycycline- and light-inducible Cre recombinase (DiLiCre). Following extensive characterization in cancer cell and organoid systems, we generate a DiLiCre mouse line, and illustrated the biological applicability of DiLiCre for light-induced mutagenesis in vivo and positional cell-tracing by intravital microscopy. These experiments illustrate how newly formed *HrasV12* mutant cells follow an unnatural move-ment towards the interfollicular dermis. Together, we develop an efficient photoactivatable Cre recombinase mouse model and illustrate how this model is a powerful genome-editing tool for biological and biomedical research.

The ability to achieve spatial control on gene expression enables researchers to design complex experiments in biological and biome-dical research. For example, the expression of oncogenes by cell-type specific promoters[1–3] and depletion of tumor suppressor genes (TSGs) by cell-type specific Cre-Lox systems[3–5] have been used to study the initial steps of cancer. However, for many genes this strategy is not possible since it leads to embryonic lethality[6] or phenotypic abnorm-alities (e.g., hypomorphism). To overcome these barriers, conditional models have been developed, in which manipulation of gene expres-sion is induced chemically after birth by e.g. tamoxifen-induced acti-vation of Cre recombinase. However, tamoxifen and other chemical inducers act systemically, they may not be delivered at sufficient concentrations to specific tissues, and they may exert toxic (off-target) effects. More recently, the advent of CRISPR based technologies have

offered new possibilities for the genomic editing of healthy adult mice at any genomic locus without the need for creating a stable mouse line. These technologies facilitate the study of the additive or synergistic effects of targeting multiple genomic loci[7,8]. However, not all organs are equally accessible to the sgRNA guide delivery, which still remains one of the main hurdles of this technology[9]. Therefore, the develop-ment of optogenetics, and especially Cre recombinases that can be activated with light, holds great promise to overcome most of the limitations of the above-described technologies to control the expression of genes in a spatial manner.

Two different types of photoactivatable Cre recombinases have been developed, based either on the photoreceptor-mediated dimer-ization of two subunits that form together an active Cre recombinase (herein, paCre)[10–13], or on the photon-mediated cleavage of a subunit

[1]Department of Molecular Pathology, Oncode Institute, Netherlands Cancer Institute, Amsterdam 1066 CX, The Netherlands. [2]Mouse Clinic for Cancer and Aging, The Netherlands Cancer Institute, Amsterdam 1066 CX, The Netherlands. [3]Division of Molecular Carcinogenesis and Oncode Institute, Netherlands Cancer Institute, Amsterdam 1066 CX, The Netherlands. [4]Cell Biophysics Group, Department of Cell Biology, The Netherlands Cancer Institute, Amsterdam, The Netherlands. [5]BioImaging Facility, The Netherlands Cancer Institute, Amsterdam, The Netherlands. [6]NKI Robotics and Screening Center and Genomics Core Facility, The Netherlands Cancer Institute, Amsterdam, The Netherlands. [7]Swammerdam Institute for Life Sciences, University of Amsterdam, Amsterdam, The Netherlands. ✉e-mail: m.vizoso.patino@gmail.com; j.v.rheenen@nki.nl

that prevents the Cre recombinase from entering the nucleus (herein, PhoCl)[14,15]. In practice, paCre based systems are difficult to apply in mouse experiments, since even the most advanced system requires long light exposure (>minutes) to induce recombinase activity. Therefore, paCre based systems have mostly been used to activate Cre recombinase in bacteria, cell lines, or cells isolated from mouse tissues[10,11,14,16,17]. The PhoCl system requires light exposure time in the range of seconds to minutes, and is therefore potentially more suited for mouse experiments. In this system, light breaks-off a domain that keeps Cre recombinase sequestered in the cytoplasm and out of the nucleus (see below for more details). Although paCre systems have been successfully applied to examine tissues ex vivo upon transient transfections in vivo[10,12,18] or by using derived cells or tissues from paCre transgenic mouse models[17,18], the use of a transgenic mouse line for e.g. lineage tracing experiments requires further improvements of the optimal tradeoff between photo-induced and background activity of the Cre recombinase.

Here, we present an advanced optogenetic Doxycycline and Light inducible Cre system (DiLiCre) in which we have optimized the ratio of photo-induced and background Cre activity. We generated a robust genetic DiLiCre mouse line and show that this mouse can be used to induce locally the expression of fluorescent lineage tracing reporters in intestinal crypts and in skin tissues. Moreover, we show the power of the DiLiCre model by locally inducing *HrasV12* mutations and following the subsequent altered behavior of those cells at different sites.

## Results

### DiLiCre, a double locked light-inducible Cre recombinase for real-time imaging

The use of photoactivatable Cre for mouse experiments requires an optimal tradeoff between light-induced Cre activity and background Cre activity. We started by reducing the background activity of currently available light-inducible Cre recombinases. Here, we adopted and modified a construct recently published by Zhang and colleagues[14], where two-flanking 405 nm-photoswitchable (green-to-red) and photo-unstable fluorophores fused to the ERT2 domain were employed to keep the Cre recombinase in the cytoplasm. We started with this construct since it has shown a good signal-to-noise ratio (dynamic range) and fast kinetics of activation[14]. To even further reduce background recombination, we cloned the original construct into a vector with a doxycycline-inducible TETon promoter to create the first two-lock photoactivatable Cre system, which we refer to as DiLiCre0.0 (Fig. 1a).

To test the recombination efficiency, we engineered HEK293T cells to harbor an inverted Cre reporter (FLEx reporter) that switches from membrane-bound mTurquoise2 to nuclear-localized RFP (herein, memMTQ2-nRFP) upon Cre recombination. The system works as follows: (i) memMTQ2 expressing cells, upon doxycycline treatment, start expressing DiLiCre0.0. (ii) DiLiCre0.0 is sequestered in the cytoplasm by binding to heat-shock protein 90 through the ERT2 domains. (iii) Once exposed to 405 nm light, PhoCl fluorophores switches from green-to-red, followed by conformational dissociation of the fluorophore and loss of fluorescence. Note that due to light scattering in tissues, it is difficult to expose cells to a well-defined amount of light, and that the green-to-red switch provides a good estimate whether cells have been sufficiently illuminated with the activation light. Due to the break of PhoCl unit during the photoswitching, the Cre recombinase gets dissociated from the ERT2 domains and translocates to the nucleus. A schematic representation of this process with special focus on how the PhoCl unit dissociates from the Cre recombinase is shown in Supplementary Fig. 1a–c. (iv) Finally, the light-mediated dissociation enables Cre recombinase to translocate to the nucleus where it recombines the memMTQ2-nRFP cassette and leading to a permanent switch from membranous blue to nuclear red (Fig. 1b).

To test this system, we exposed HEK293T cells stably expressing DiLiCre0.0 and the Cre reporter memMTQ2-nRFP to doxycycline for 24hrs. As expected, doxycycline induced the expression of green DiLiCre0.0 which was located in the cytoplasm (Supplementary Fig. 2a, b). Next, we exposed some of the clusters of cells to violet light (see blue boxed areas in Supplementary Fig. 2a, b), which induced the DiLiCre0.0 to switch from green to red (see white boxed areas in Supplementary Fig. 2a, b and zoom-ins in Supplementary Fig. 2c). Although the red DiLiCre0.0 ceased over time suggesting for the breakage of the ERT2 domains from the recombinase, we did not observe any Cre recombinase activity that resulted in the membranous blue to nuclear red switch (Supplementary Fig. 2d, e). This is not caused by the inability of DiLiCre0.0 to enter the nucleus, since treatment of cells with 4-OH-tamoxifen, which binds to the ERT2 domains, induces a clear translocation of DiLiCre0.0 to the nucleus (Supplementary Fig. 2f–h) but no recombination upon photoactivation (Supplementary Fig. 2i). Moreover, transfection of the cells with a regular pCMV-Cre vector induced a permanent switch from membranous blue to nuclear red further suggesting that the lack of recombination of the memMTQ2-nRFP cassette by light-activated DiLiCre0.0 is not caused by a damaged reporter (Supplementary Fig. 2j). From these experiments, we conclude that exposure of DiLiCre0.0 to light does not induce recombination of the Cre reporter.

To repair the light-activation of DiLiCre, we removed the N-terminal ERT2-PhoCl repeat leading to DiLiCre1.0 and stably expressed this construct in HEK293T and C26 cells (Fig. 1a). Upon doxycycline and violet light exposure (in the boxed area), DiLiCre1.0 gets expressed and switches from green to red (Fig. 1c). Within a few minutes, the cytoplasmic red fluorescence of DiLiCre1.0 disappears as expected when the ERT2 domain dissociates from the Cre recombinase. Importantly, and in contrast to experiments with the 0.0 version, light-activation of DiLiCre1.0 leads to a permanent switch from membranous blue to nuclear red, showing light-activation of the Cre recombinase (Fig. 1d, Supplementary Movie 1 and 2, and Supplementary Fig. 3a). Moreover, we quantified the decay of memMTQ2 in recombinant nRFP positive cells and confirmed that memMTQ2 levels drop significantly upon recombination (Supplementary Fig. 3b, c). In the absence of doxycycline, cells did not express DiLiCre1.0 and therefore transient photoconversion (green-to-red) was not observed even if cells were illuminated with 405 nm laser light (Supplementary Fig. 3d). From all these data we conclude that DiLiCre1.0 is a double locked light-activatable Cre recombinase that can be activated by short periods of low-intensity violet light.

### DiLiCre2.0: reducing the background recombination whilst maintaining the efficiency of light induction

Although the removal of one of the two flanking ERT2 domains was required to obtain Cre activity upon light, this modification also led to a theoretical decrease in cytoplasmic sequestering with subsequent increase in background Cre activity. Indeed, we detected leaks on membranous blue to nuclear red switch in control wells treated with doxycycline but not with light (Supplementary Fig. 3e–g and Supplementary Movie 3). To further optimize the ratio of light-induced and background Cre activity, we designed an experimental setup in which we can compare Cre recombinase activity for different variations of DiLiCre in a semi-quantitative manner. We seeded cells in 24-well plates and exposed all wells simultaneously with a home-built 3D printed 24-array LED box (Supplementary Fig. 3h), followed by analysis by flow cytometry. First, we confirmed by flow cytometry that DiLiCre1.0 has a higher background recombination activity than DiLiCre0.0 upon doxycycline treatment (non-light exposure) (Supplementary Fig. 3i). Next, we used this approach to test whether the expression level of DiLiCre1.0 has an influence on the ratio of light-induced and background activity of the Cre recombinase. We generated cells with low, intermediate and high expression levels of

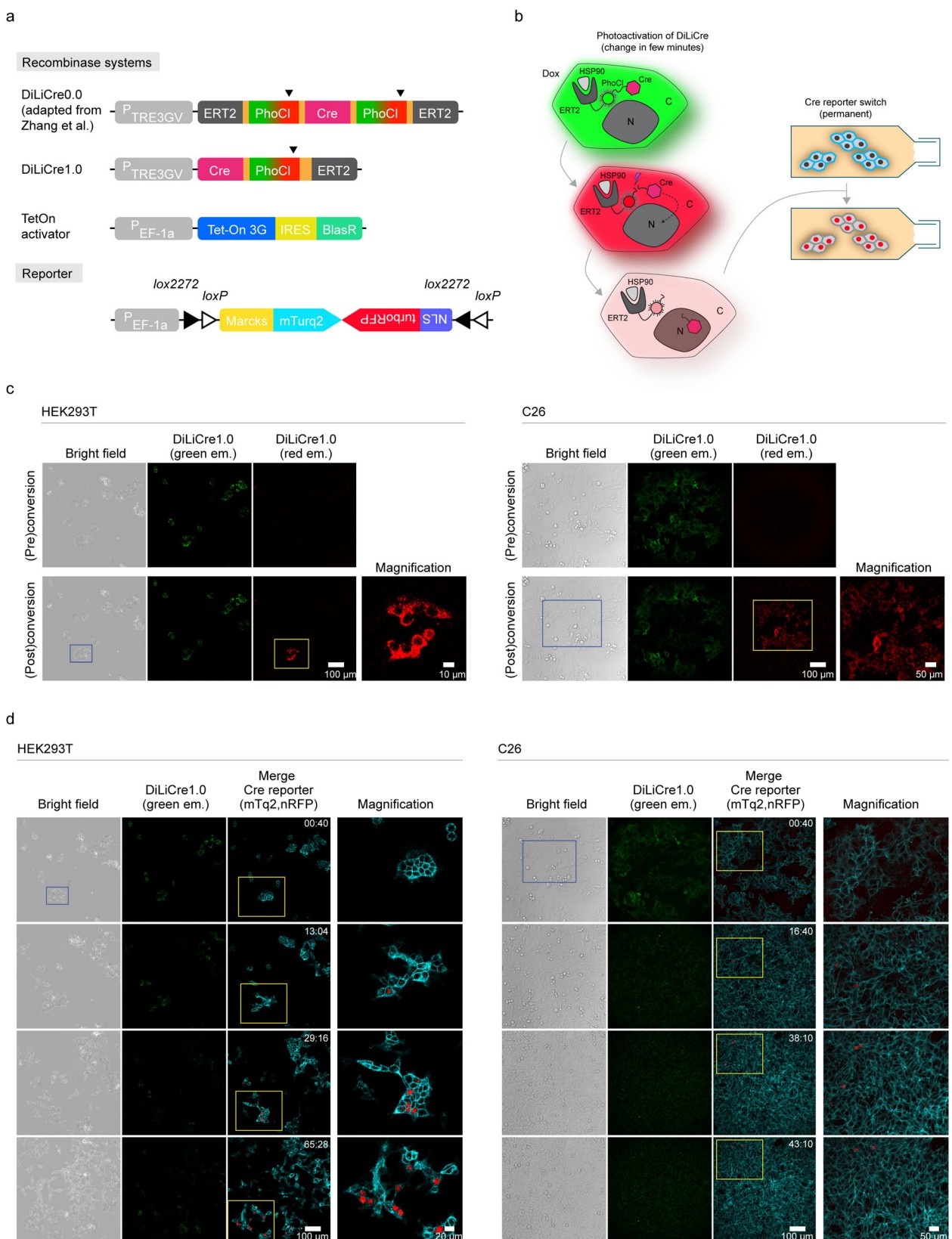

DiLiCre1.0, and exposed the cells to different levels of doxycycline. The efficiency of both background and light-induced Cre recombinase activity is positively correlated with DiLiCre1.0 levels (Supplementary Fig. 3j). However, from this analysis it is also clear that removal of one of the ERT2 domains in DiLiCre0.0 leads to a significant amount of background activity (Supplementary Fig. 3j).

A light-activatable Cre recombinase is most useful if the background recombination levels can be kept low. To decrease the background activity, we further modified the DiLiCre1.0 by fusing a flexible linker and a second ERT2 domain onto the C-terminus of the first ERT2 domain (Fig. 2a). This modified version, that we refer to as DiLiCre2.0, has two ERT2 domains to prevent transportation to the nucleus but

**Fig. 1 | DiLiCre1.0 system to photo-induce Cre activity. a** Schematic representation of the various photoactivatable Cre recombinase systems: (i) DiLiCre0.0 (adapted from Zhang et al. (2016)), containing the tetracycline response element promoter (TRE3GV) and two flanking PhoCl-ERT2 tandem repeats, and (ii) DiLiCre1.0 construct lacking the N-terminal ERT2-PhoCl tandem repeat. Moreover, schematic representations of the transactivator of the TETon system and the Cre reporter are shown. The Cre reporter is a FLEx vector containing the membrane mTurquoise2 (non-induced) and the nuclear RFP fluorophores (induced). Arrowheads indicate the light-breaking points of the PhoCl fluorophore. **b** Schematic representation of the mode of action of DiLiCre that recombines the memMTQ2-nRFP reporter. All other DiLiCre systems work in a similar fashion. In brief, doxycycline treated cells express DiLiCre which is sequestered in the cytoplasm by binding to heat-shock protein 90 through the ERT2 domain. As a consequence, the green light emitted from the DiLiCre fluorescent molecule (PhoCl) is observed in the cytoplasm. Upon 405 nm light exposure, PhoCl switches from green-to-red fluorescence, followed by conformational dissociation and cleavage of the

fluorophore and subsequently loss of fluorescence. Due to the break, the region containing Cre recombinase dissociates from the ERT2 domain and translocates to the nucleus where it recombines the memMTQ2-nRFP cassette leading to a permanent switch from membranous blue to nuclear red fluorescent. **c** Representative confocal images of HEK293T (left images) and C26 (right images) cells stably expressing DiLiCre1.0 and memMTQ2-nRFP reporter, before and after photoactivation. The photoactivated region is boxed by blue lines and is mediated by single pulse of laser light at 405 nm (HEK293T: 1.6 mW/mm² for 45-90 s pulse; C26: 7.25 mW/mm² for 45 s). A zoom image of the yellow boxed area is shown at the right of each image. This experiment was independently repeated $n = 10$ times with similar results. **d** Time-lapse confocal series of HEK293T (left images) and C26 (right images) cells that have been photoactivated in panel **c**. A zoom image of the yellow boxed area is shown at the right of each image. Experiments were independently repeated $n = 2$ times with similar results. The numbers inside the images represent hours and minutes (hh:mm).

only one light-inducible cleavage site. To test whether this modification of DiLiCre improves the cytoplasmic retention whilst leaving the light-inducible nuclear translocation intact, we isolated cytoplasmic and nuclear fractions of cells. Importantly, no contamination of the cytoplasmic heat shock protein 90 and the nuclear protein Histone 3 was observed at the experimental detection level in the nuclear and cytoplasmic fractions, respectively, illustrating successful separation of the nuclear and cytoplasmic fractions (Fig. 2b). When blotting for Cre, we made several important observations. First, as we expected, DiLiCre2.0 had a higher molecular weight than DiLiCre1.0 (137 kDa versus 102 kDa, upper panel Fig. 2b). Second, the amount of non-light-activated DiLiCre that leaks into the nucleus was lower for the 2.0 than 1.0 version (the upper band in lane 9 versus 3 of the upper and middle panels of Fig. 2b and left plot in Fig. 2c). Third, violet light exposure induced an efficient cleavage and dissociation of ERT domain(s) from the two DiLiCre versions (DiLiCre1.0: lanes 3–4, and DiLiCre2.0: lanes 9–10, middle panel, Fig. 2b). Fourth, the translocation to the nucleus upon this light-induced cleavage was stronger for the 2.0 than the 1.0 version (middle panel: lanes 10 and 11 versus 4 and 5, lower bands, Fig. 2b and right plot in Fig. 2c). Five, the expression of DiLiCre proteins is extinguished 24 hours after doxycycline treatment (DiLiCre1.0: lane 6, and DiLiCre2.0: lane 12). To test whether these differential characteristics also leads to differential ratios of light-induced and background Cre activity, we transfected cells that express a red nuclear Cre reporter with the two DiLiCre versions. In this reporter, the absence of red fluorescence indicates Cre activity. Flow cytometry confirmed a better ratio of light-induced over background Cre recombinase activity in the 2.0 compared to the 1.0 version in all conditions tested (e.g. for high expressing cells: DiLiCre1.0 up to 1.3-fold and DiLiCre2.0 up to 2.1-fold, Fig. 2d and Supplementary Fig. 4a, b). Overall, we can conclude that DiLiCre2.0 system performs better than DiLiCre1.0 in respect to cytoplasmic retention and nuclear leakage values resulting in an optimized ratio of light-induced and background Cre activity.

## DiLiCre2.0, a light-inducible Cre mouse model to activate Cre activity by light in vivo

To perform in vivo experiments, we generated C57BL/6J embryonic stem cells (mESC) in which DiLiCre2.0 was integrated in a locus just 3' to the *Col1a1* gene. We successfully photoconverted the mESC by 405 nm light (Supplementary Fig. 5a) which subsequently led to the expression of transiently transduced Cre reporter (Supplementary Fig. 5b). From these mESC we generated a DiLiCre2.0 mouse. Next, we treated 8 week old animals with doxycycline and evaluated the expression of DiLiCre2.0 in various organs. Most of the organs shown a homogeneous cytoplasmic doxycycline dependent expression of DiLiCre2.0, although some patchiness was observed in organs like the mammary glands and the lungs among different cell types (Supplementary Fig. 5c).

To test whether this DiLiCre mouse line can be used for inducing lineage tracing markers, we crossed these animals with R26-Confetti mice, which contains a stochastic multicolor Cre recombinase reporter of multiple fluorescent proteins from a single genomic locus[19] (Fig. 3a). From these animals, we derived small intestine organoids. We exposed the organoids to different levels of doxycycline and subsequently to light (Fig. 3b). Importantly, we observed high efficiency of light-induced recombination whilst a background signal remained nearly absent (Fig. 3b and Supplementary Fig. 6a). In addition to flow cytometry, we imaged the organoids for 40 h by confocal microscopy (Fig. 3c). In this experiment, we tested two types of violet-light sources to activate DiLiCre: exposure of the whole well with our custom 3D printed 24-array LED box, or exposure of a single confocal z-plane (Supplementary Fig. 6b). In addition to the non-specific auto-fluorescence in the lumen, light-activation of DiLiCre2.0 led to the appearance of individual confetti labeled cells. Although a few confetti labeled cells also appeared in the doxycycline treated and non-light exposed organoids, light exposure substantially increased the number of confetti labeled cells (Fig. 3c and Supplementary Fig. 6c). To quantify this difference, we used a machine-learning approach to identify fluorescent cells (see materials and methods), and quantified the recombination activity by measuring the appearance of (C/G/Y/R) FP fluorescence. In line with the flow cytometry data, we observed many more confetti signal in organoids that were exposed to both violet-light and doxycycline than in organoids expose to only violet-light or doxycycline (Fig. 3d).

Next, we tested the light induction of Cre activity in living mice. After surgical implantation of an abdominal imaging window, DiLiCre2.0 mice were treated with doxycycline. Through the imaging window, we performed multi-day intravital microscopy of the same regions of interest of the small intestine (Fig. 4a). Regions with ~1–4 intestinal crypts were exposed to violet light (i.e. spatially activated) to induce temporal green-to-red switch (Fig. 4b and Supplementary Fig. 7a), leading subsequently to the breakage of the ERT domains and loss of the red signal. Similar to the in vitro experiments, we observed recombination events of the confetti reporter appearing in the light-exposed regions 24 h post-activation (Fig. 4c and Supplementary Fig. 7b). This recombination occurs along the crypt-lumen axis indicative for the activation of DiLiCre2.0 (Fig. 4d). The quantification of confetti cells and fluorescence intensity indicated much more prominent induction in the violet-light exposed regions than in the control regions (Fig. 4e and Supplementary Fig. 7c–e) suggesting optimized ratio of light-induced and background Cre activity. Similar low background recombination levels were found in animals that were treated with doxycycline but not be exposed to light (Supplementary Fig. 8a–d). Overall, these results show that we have successfully generated a mouse line that expresses DiLiCre2.0 that can be activated by short-term exposure to violet light.

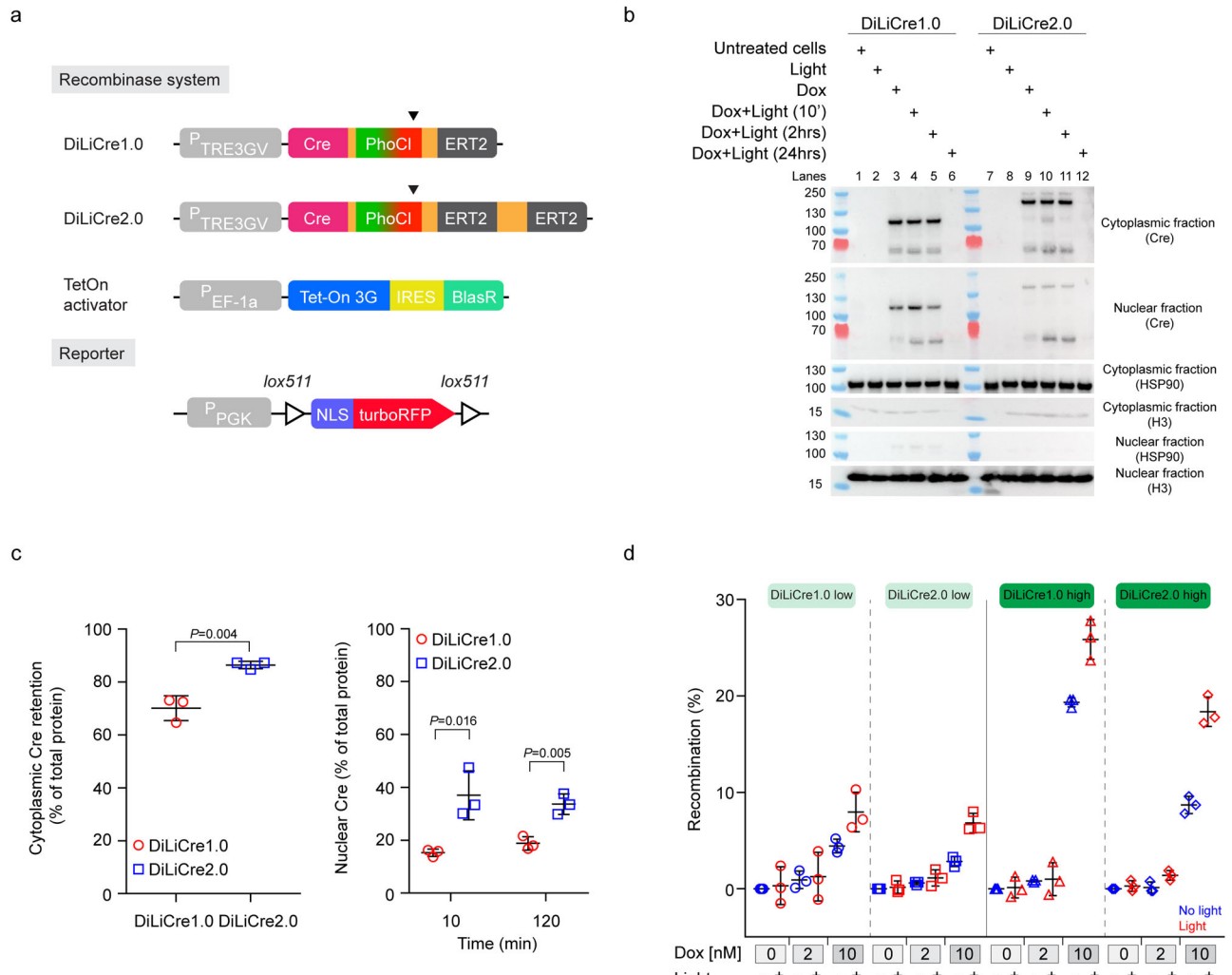

**Fig. 2 | Modifications in DiLiCre2.0 reduce the background recombination whilst maintaining the efficiency of light induction. a** Schematic representation of DiLiCre1.0 and DiLiCre2.0 constructs and the floxed nuclear-RFP cassette. Arrowheads the light-breaking points of the PhoCl fluorophore. **b** Representative nuclear-cytoplasmic protein fractionation time course experiment carried out in DiLiCre1.0 and DiLiCre2.0 HEK293T cells. Four different conditions were tested: untreated cells, cells exposed to doxycycline or light, and combined treatment (10 nM dox and 0.5 mW/mm² 90-s-light pulse). Three time points were compared (10 min, 2 h, and 24 h). $n = 3$ biologically independent experiments were performed with similar results. Protein molecular weights (kDa) are annotated on the left side of the scans. **c** Protein densitometries from the nuclear-cytoplasmic assays. Plots represent the percentage of total Cre retained in the cytoplasm (left graph) or

delivered in the nucleus (right graph). All percentages were normalized by the total Cre protein content. Data represents the mean ± SD from $n = 3$ biologically independent experiments. $P$ values were calculated by a two-tailed $t$-test. **d** FACS quantification of the light-mediated recombination in HEK293T cells stably expressing the nuclear RFP reporter and sorted for low and high expression of DiLiCre1.0 or 2.0 constructs. All sorted populations were tested for two different concentrations of doxycycline (2 or 10 nM). For each treatment, cells were both exposed and non-exposed to light (0.5 mW/mm² 30 s pulses every 4 h [x3]). Control values (untreated cells) were subtracted. Data represents the mean ± SD from $n = 3$ biologically independent experiments. Source data are provided as a Source Data file.

## Optogenetic induction of oncogenic cell transformations in vivo

Finally, we tested whether the DiLiCre2.0 mouse can be used for biological experiments by light-inducing an oncogenic mutation in individual cells. We crossed the DiLiCre2.0 mice with *DMT1-Stop-HrasV12-eGFP* animals, which contained a floxable and excisable Lox-Stop-Lox cassette preventing *HrasV12* oncogenic mutation to be expressed unless Cre is activated (Fig. 5a). Upon Cre recombinase activity, new transformed cells emerge that are positive for eGFP. Since this particular mutation is relevant for basal and squamous cell carcinoma of the skin[20,21] (present in 66% and 18% of mutated samples, respectively, COSMIC database), we decided to activate DiLiCre2.0 in the skin of the ear (Fig. 5b). We used the vasculature in auto-fluorescent wide field images and the hair follicles (HF) in the second harmonic generation (SHG) multiphoton images (i.e. collagen type I) as a roadmap to revisit the same positions over multiple imaging sessions. The day after

doxycycline treatment, we observed non-specific background recombination in a small fraction of the large suprabasal keratinocytes. These fluorescent cells quickly evanesced in the subsequent days as a consequence of the epidermis renewal (Supplementary Movie 4, corresponding to Fig. 5c). To specifically induce mutant cells by light, we exposed four-to-five regions with violet light from the top to the bottom of the HF leading to the green-to-red conversion and subsequent loss of fluorescence and activation of DiLiCre2.0 (Supplementary Fig. 9a). Forty-eight hours after activation of DiLiCre2.0, HrasV12-eGFP transforming cells started appearing in the violet-light exposed regions of interest (mostly concentrated within the HFs regions; Fig. 5c and Supplementary Movie 4). Orthogonal views from targeted-hair follicles taken 48 h post-illumination validated the presence of mutant cells within and along the hair follicles (Supplementary Fig. 9b). The amount of fluorescence of HraV12 is one order of magnitude larger

a

b

c

d

| YFP/RFP | Dox-,light | Dox+, non-light |
|---|---|---|
| Dox-, light | | 0,999/0,957 |
| Dox+, non-light | 0,999/0,957 | |
| Z-plane | <0,001/<0,001 | <0,001/<0,001 |
| WFI | <0,001/<0,001 | <0,001/0,001 |

than the fluorescence of DiLiCre2.0. Quantification of HrasV12-GFP throughout the skin revealed significant differences in induction of the oncogene in violet-light exposed compared to non-light exposed control (Fig. 5d and Supplementary Fig. 9c, d). Similar low background recombination levels were found in animals that were treated with doxycycline but not be exposed to light (Supplementary Fig. 10a–c and Supplementary Movie 5).

Finally, we attempted to light-activate DiLiCre2.0 at a specific site by exposing the cells at the basal layer to violet light. We detected a thin layer of small rounded basal cells expressing DiLiCre2.0 at the interfollicular epidermis by second harmonic generation (SHG)

imaging of fibrillar collagen type I (Fig. 6a). Orthogonal sections of the z-stacks at the illuminated areas revealed that the first days after light-induction, HrasV12-eGFP positive cells appear on top of the collagen-I layer (Fig. 6b). Strikingly, revisiting the same areas the following days revealed that the dermis gets invaginated (seen as deformation of fibrillar collagen-I) at the regions of the transformed cells at $5.8 \pm 2.5$ days post light induction (Fig. 6b and Supplementary Data 1). Interestingly, the next days, HrasV12-eGFP cells in the basal layer are lost whilst those mutant cells appeared in the dermis (inside the region with collagen type I) (Fig. 6b). Total intensity measurements of eGFP and SHGs along the z-axis confirmed the relocation of the HrasV12-

**Fig. 3 | DiLiCre2.0 small intestine organoids recombine ex vivo with high efficiency. a** Schematic representation of the design to generate the DiLiCre2.0 mice and the breeding with Rosa26-Confetti mice. Recombination outputs of the Confetti cassette are annotated. The Confetti cassette contains a LoxP-flanked NeoR-cassette serving as transcriptional roadblock (NeoS) which prevents any fluorophore of the array to be expressed unless Cre becomes active. **b** Graphical scheme of the FACS experiment setup for the evaluation of the genetic recombination in small intestine derived organoids from the DiLiCre2.0;R26-Confetti mice. Quantification of FACS data (left graph) and dynamic range (right graph) derived from the experiments in panel **b** (see representative flow cytometry plots in Supplementary Fig. 6a). Data represents mean ± SD from $n = 3$ biologically independent experiments. **c** The upper image shows the experimental timeline. In brief, organoids

were treated with doxycycline (50 nM) and/or exposed to whole field illumination (WFI, 405 nm LED light 0.5 mW/mm$^2$ 30 s pulse) or z-plane confocal illumination (imaged with 1.87 mW laser output power for 30 s). The lower panel consist of representative maximum projection images at different time points. Blue, green, yellow, and red color code represents mCFP, nGFP, cYFP, and cRFP, respectively. This experiment was independently repeated $n = 3$ times with similar results. **d** Schematic summary of the workflow to quantify the light-mediated recombination in small intestine organoids. Plots represent the CFP/GFP/YFP or RFP total intensities divided by the organoid size. Data represents the mean ± SD from $n = 3$ biologically independent experiments. Entire curves were compared statistically and the $P$ values were determined by permutation test adapted from Elso et al. (2004). Source data are provided as a Source Data file.

eGFP cells (Fig. 6c and Supplementary Fig. 11a, b). The change in location of the HrasV12-eGFP cells either means that these cells have invaded the dermis or have a differential expansion at the basal layer and the dermis. Overall, these experiments demonstrate that our DiLiCre2.0 mouse model can be used to initiate fluorescent oncogenic mutant cells and study their fate over time.

## Discussion

In this study, we aimed to develop a photoactivatable Cre-expressing mouse model optimized for in vivo lineage tracing experiments. Over the years, multiple optogenetic systems based on light-inducible Cre recombinases, have been developed with different characteristics, but none were optimized for in vivo use. Our data illustrates that the characteristics of DiLiCre2.0 are optimal for in vivo experiments because: (1) the system does not cause a (lethal) phenotype in mice, (2) the background activity is minimized since DiLiCre2.0 is only expressed upon doxycycline, (3) the green fluorescence of DiLiCre2.0 is only expressed temporally and does not interfere with any green lineage tracing mouse models, (4) for each organ/tissue of interest, the ratio of activation and background activity can be optimized by tuning the expression level of DiLiCre2.0 using various concentration of doxycycline, (5) the optimal exposure to the activating light (both in intensity and time) can be optimized by the green-to-red conversion of DiLiCre2.0. This is particularly important in vivo since light scattering in tissues renders light intensity in the focal plane unknown. Lastly, (6) the green-to-red switch illustrates which cells have been exposed to light, which is critical for in vivo experiments where tissue movements during imaging (and thus during activation) are common.

Prior photoactivatable Cre systems have mainly been optimized and used for the ex vivo analysis of cells or tissues[10,12,17,18]. The background activity of previous reported systems is seemly low due to (i) the use of enzymatic reactions (e.g., luciferase assays) as a readout of Cre recombinase activity or (ii) the short-term (<24 h) follow up time after activation[10,12,18,22]. This is likely leading to an underestimation of the background recombination levels when applying these systems for in vivo applications, hence explaining the lack of any reports to use these systems for e.g. positional cell-tracing experiments. Indeed, extensive comparative analysis of existing photoactivatable systems often challenges the original reported light-inducible and background activity (Duplus-Bottin et al.; Supplementary Fig. 12a, b). Moreover, optimized in vitro characteristics does not necessarily provide any relevant information about their in vivo performance. For example, we noted that the PA-Cre2.0 had a great light-inducible and background activity ratio, but that this construct is either silenced or not stably integrated in HEK293T cells (Supplementary Fig. 12c, d). In order to truthfully compare in vivo performance of DiLiCre2.0 to other systems, future benchmark experiments need to be performed in mice, which require future effort to generate mouse models of other systems.

Similar to any Cre recombinase system, we found that both the light-induced and background activity of DiLiCre2.0 depends on three factors: (i) expression levels of the DiLiCre2.0, (ii) recombination

efficiency of the reporter, and (iii) exposure time and intensity of the activating violet light. Importantly, for most in vivo experiments, exposure times to violet light should not exceed the minute range. Indeed, we observe significant light-induced Cre recombinase activity of DiLiCre2.0 with short exposure times (<1 min). Yet, it is important to emphasize that for each experimental setting and for every Cre reporter system, several factors need to be optimized including the expression levels of DiLiCre2.0 by different amounts doxycycline and the exposure time and intensity of activating violet light. Future optimization of these setting is required for each of the organs that express the DiLiCre2.0 system (i.e. breast, liver and lungs, ovary, kidney). Although background recombination cannot be fully avoided, it can be minimized by the above optimizations to reach a baseline that would not substantially interfere with the experimental setting or aim. This was shown when revisiting the full z-stack images of our in vivo experiment in the skin, the most environmentally exposed tissue and therefore the one more susceptible to be non-specifically activated by light.

The optimized tradeoff between light-activation and background activity allow us to perform in vivo induction of DiLiCre2.0 and positional cell-tracing experiments. Similar to many photoswitchable proteins[23,24], we observed that the PhoCl unit of DiLiCre2.0 can only be photoswitched with single-photon violet (405 nm) light but not with multiphoton illumination (805-820 nm) (Supplementary Fig. 13). The photoactivation of DiLiCre2.0 enables us to discover that targeted basal layer cells of the skin initiated tissue invagination reaching the collagen dermal layer few days after *HrasV12* oncogene was activated. Interestingly, we observed HrasV12-GFP cells in the dermal layer in the following days, whilst the natural movement of cells from the basal layer is towards the epidermis during skin regeneration. This means either that HrasV12-GFP cells are lost in the basal layer and simultaneously amplified in the dermal layer, or that the HrasV12 expression enables basal cells to move in the opposite direction into the dermal layer. Interestingly, the latter possibility is in line with recent observations from the Greco lab of aberrant behavior of HrasV12 mutant cells sitting outside the cycling follicular niche[2]. Further work will be needed to confirm these results and to find the molecular mechanisms behind this phenotype. Yet, the HrasV12 positional cell-tracing data illustrates that our technological improvements of DiLiCre2.0 now enables us to perform in vivo positional cell-tracing experiments, which opens advanced genome-editing tools and approaches for biological and biomedical research.

## Methods

### Ethical statement

All animals and experiments were conducted under the guidelines by the Animal Ethics Committee of the Netherlands Cancer Institute and performed in accordance with institutional, national and European guidelines for Animal Care and Use. All the animal protocols and surgery and imaging procedures were reviewed and approved by the Animal Care Committee of the Netherlands Cancer Institute (codes: 9.2.9867, 9.2.9917).

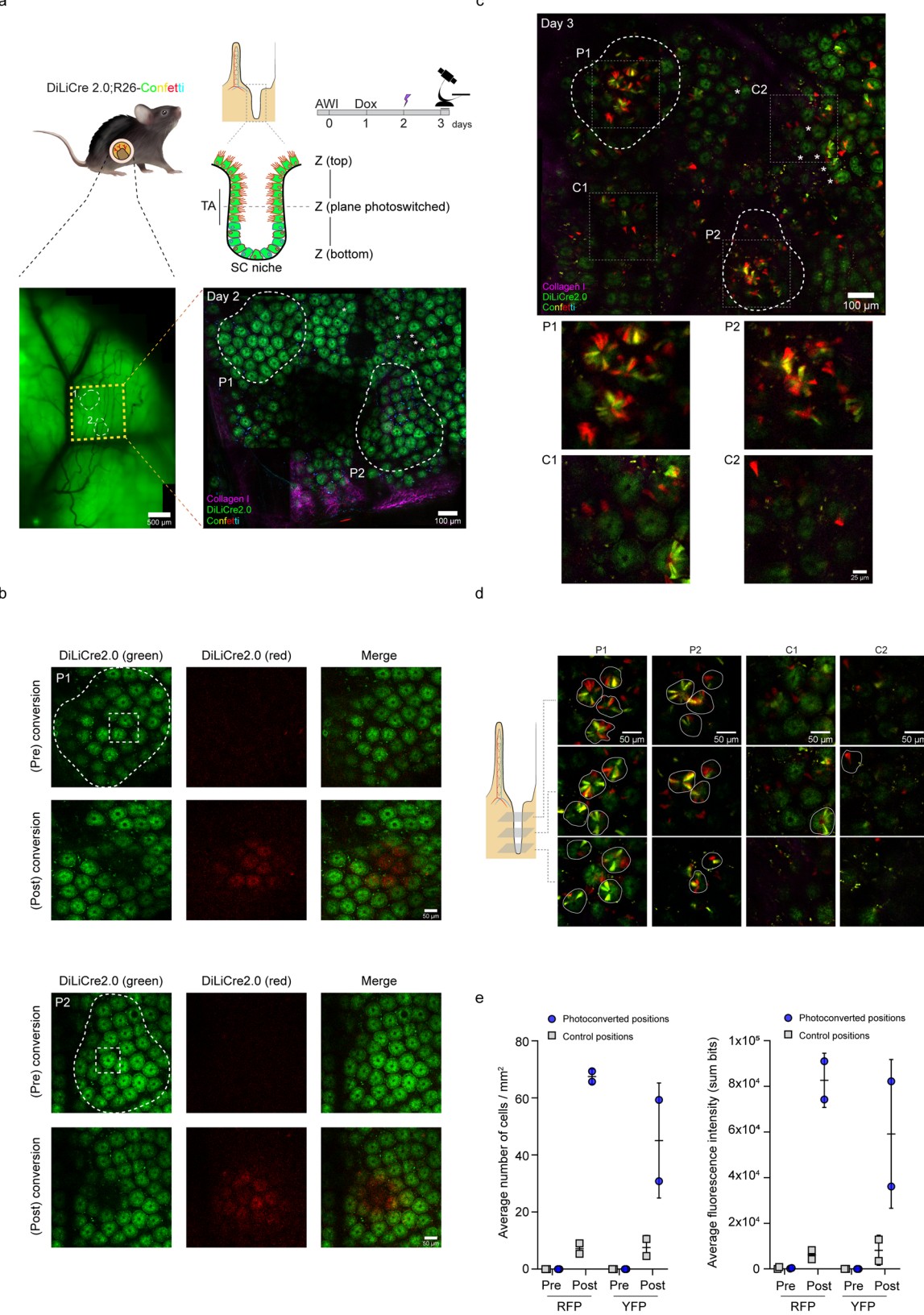

### Constructs and Cre reporters

First, we developed a third generation lentiviral doxycycline-inducible photoactivatable Cre system (vector 1 or DiLiCre0.0) by adopting and modifying the PhoCl construct14 (Addgene #87694). Cloning was performed using a third generation lentiviral plasmid as a backbone (#71666) digested with NheI-EcoRI and ligated to the following DNA

fragments: TREGV 3rd generation promoter (amplified by PCR using Addgene #85556 vector as a template and digested with NheI-XmaI), PhoCl-ERT2-Cre-ERT2-PhoCl sequence (#87694) digested with XmaI-NotI, and NotI-EcoRI adapter. A lentiviral UbC-blasticidine-P2A-TETon-3G transactivator vector (vector 2) was synthesized separately using the #71666 vector backbone digested with NheI-EcoRI and

**Fig. 4 | DiLiCre2.0 mouse model to perform positional-cell tracing in the intestine in vivo. a** Schematic representation of the mouse model and the experimental setup. Left image shows a widefield tissue overview of the small intestine, highlighting the area scanned by two-photon microscopy (yellow boxed square). The right image shows the two-photon image corresponding to the yellow boxed area on the left, where individual intestinal crypts can be identified. This image also contains the two regions of interest (ROIs) (P1 and P2) that were subsequently subjected to photoactivation (white dotted circles). White asterisks are displayed to help the reader to recognize the same structures in the following panels. This experiment was independently repeated $n = 3$ times with similar results. **b** Magnifications of the two ROIs from panel **a** are depicted. Within each ROI, regions boxed in white dotted squares are subsequently exposed to 405 nm laser light (imaged with $7.95 \pm 2.19$ mW laser output power for 45 s). Images before and after photoconversion are shown. A Gaussian-blur filter was applied for visualization purposes. **c** A longitudinal two-photon overview of the same intestinal area shown in panel **a** (yellow square) 24 h post-photoconversion is shown. The two ROI regions described in panel **a** (P1 and P2) and the two zoom-in photoconverted areas depicted in panel **b** are indicated. Control regions are also annotated (C1 and C2). Control positions were selected randomly at least 150 μm away from the photoconverted areas. Detailed magnifications from both photoconverted and control regions are shown below. **d** Representative single z-planes of two-photon images along the submucosa and the lumen axis one day after photoconversion. Clusters of RFP or YFP "wedge" cells within the intestinal crypts are annotated using white dotted lines. Light exposed positions (P) and control regions (C) are shown. **e** Quantification of RFP and YFP cell numbers and fluorescence intensities pre- and post-photoconversion (-24 h post-conversion). Measurements were performed in the photoconverted and non-photoconverted areas. Same square dimensions were used for all measured positions ($225 \times 225$ μm). Control positions were selected randomly at least 150 μm away from the photoconverted areas. Data represents mean $\pm$ SD. $n = 3$ biologically independent experiments/mice were performed with similar results. Source data are provided as a Source Data file.

ligated to the following DNA fragments: UbC promoter and blasticidine cassette (from in-house vectors) amplified and fused by PCR using 40–50 bp internal overlapping primers and one external reverse primer containing a P2A linker, and a TETon-3G fragment amplified from the Addgene vector #96963 using a forward primer containing a P2A sequence so it could be fused to the UbC-Blasticidin-P2A sequence during a last round PCR amplification with NheI and EcoRI external primers.

A simplified version of vector 1, named DiLiCre1.0, was synthesized as follows. From construct PhoCl (#87694), the sequence Cre-PhoCl-ERT2 was amplified. Cre sequence from the original construct lacked the canonical first 16 aminoacids (MANLLTVHQNLPALPV), so we restored them. The synthesized fragment was ligated into vector 1 upon PacI and EcoRI digestion. An optimized version of vector DiLiCre1.0, named DiLiCre2.0, was synthesized as follows. Using DiLiCre1.0 as a template, we amplified Cre-PhoCl-ERT2 sequence to generate a new fragment flanked by PacI and AscI restriction sites. An additional ERT2 domain was synthetized by PCR with AscI and EcoRI ends and one 5′-GSGSGGG flexible linker. Upon digestion, the two fragments were ligated into vector 1 upon PacI and EcoRI digestion. Finally, Cre-PhoCl-ERT2(x2) fragment was jumped into the PiggyBac XLone vector system[25]. To this end, XLone plasmid backbone was PCR and Cre-PhoCl-ERT2(x2) cassette ligated upon NheI-NotI digestion.

DiLiCre2.0 was also optimized for mESC transfection (DiLiCre2.0-mESCs), by generating an all-in-one plasmid containing the F3-TRE3GV-Cre-PhoCl-ERT2(x2)-β-globin-Intron-bGH-polyA-CAG-TetOn3G-bGH-polyA-FRT sequence. First, F3-FRT acceptor plasmid (in-house) was digested with PacI-EcoRI and CAG promoter (from in-house vector) was cloned into it using the same restriction sites. Then, TETon-3G fragment was amplified by PCR, digested, and ligated into the new plasmid using EcoRI-SacI sites. The new plasmid was digested with PacI-MluI to add a DNA adapter to include the NheI restriction enzyme. Finally, the plasmid was digested with NheI-MluI and two fragments were ligated: NheI-Cre-PhoCl-ERT2(x2)-NotI (obtained from DiLiCre2.0 and dephosphorilated) and NotI-β-globin-intron-bGH-polyA-MluI (amplified from an in-house vector).

In this study, we generated two Cre reporter plasmids. First, the lentiviral EF1α-Lox2272-LoxP-MARKS-mTurquoise2-NLS-turboRFP-Lox2272-LoxP double-floxed inverted open reading frame reporter. Using #37120 addgene plasmid as a backbone (cut AscI-NheI), we ligated the following fragments: first, we PCR the membrane bound sequence (MARKS, Addgene #17787), the mTurquoise2 sequence (+stop codon, from in-house plasmid), and the nuclear NLS-turboRFP sequence (+stop codon) from #89587 addgene vector. MARKS was fused by PCR to mTurquoise2 using forward overlapping regions and NLS-turboRFP was fused to mTurquoise2 in a reversed orientation. The fusion product was ligated to the lentiviral backbone using the AscI-NheI restriction enzyme sites. The second lentiviral Cre reporter was the PGK-Lox511-NLS-turboRFP-Lox511 excision reporter. The PGK promoter (in-house plasmid) was amplified and fused to the NLS-turboRFP sequence (#89587). The resulting product was digested with EcoRI and NsiI restriction enzymes, and DNA fragment ligated between the Lox511 sites from Addgene vector #11586.

Lentiviral infections were performed as follows. For viral particle production, HEK293T cells were seeded at 80 % confluence the day before transfection in one 10 cm dish. Next day, after refreshing media, 8 μg of target plasmid was mixed with 2.25 μg VSV-G, 3.125 μg pMDLg/pRRE, and 1.56 μg REV plasmids in 1 ml of Optimem media. 40 μls of lipofectamine were resuspended in 1 ml of Optimem and then mixed with the plasmid suspension. Transfected cells were incubated with the plasmid/lipofectamine cocktail overnight. Then, the media was refreshed and viruses obtained after 48 h. For transduction, viral particles were concentrated using Lenti-X Concentrator (Takara, cat. no 631232) and tittered by qPCR (Abmgood, cat. no. LV900) following manufacturer instructions. Multiplicity-of-infection (MOIs) values of 10–100 were used in this study.

DiLiCre2.0 plasmid transfections were performed as follows. 24 h before transfection cells were seeded in T75 flasks so next day 80–90% of cell confluence was reached. Control cells were lipofected (Lipofectamine2000, Fisher Scientific, cat. no. 11668030) with 1 μg of DiLiCre2.0 plasmid and targeting cells with 1 μg of DiLiCre2.0 plasmids and 1 μg of pT2-PB-L3-ERT2-mCherry plasmid containing the tamoxifen inducible transposase[26]. Plasmids were resuspended in 500 μls of Optimem media and mixed with 10 μls of Lipofectamine2000 and another 500 μls of fresh Optimem. After overnight incubation, media was refreshed adding tamoxifen (1 μM) and 24 h later, tamoxifen was withdrawn.

All PCR and fusion PCR were performed using high fidelity Fusion polymerase (reference) and ligation steps were performed using T4 DNA ligase (NEB, M0202) with overnight incubations at 4 degrees.

## FACS sorting and analysis

To generate the in vitro models containing a paCre inducible system, cells were first transduced with the transactivator vector (vector 2) and selected with blasticidine (10 μg/ml) for 1 week. Afterwards we transduced or transfected with the corresponding paCre system. Cell sorting was performed to recover cell populations with homogeneous levels of DiLiCre expression in Aria Fusion (BD Bioscience) upon 24 hours of doxycycline treatment (0.5–1 μM). Once expanded, cells were transduced with the corresponding Cre reporter and 5 days later sorted again for reporter enrichment. Cells to be sorted were resuspended in FACS buffer containing 2 mM of

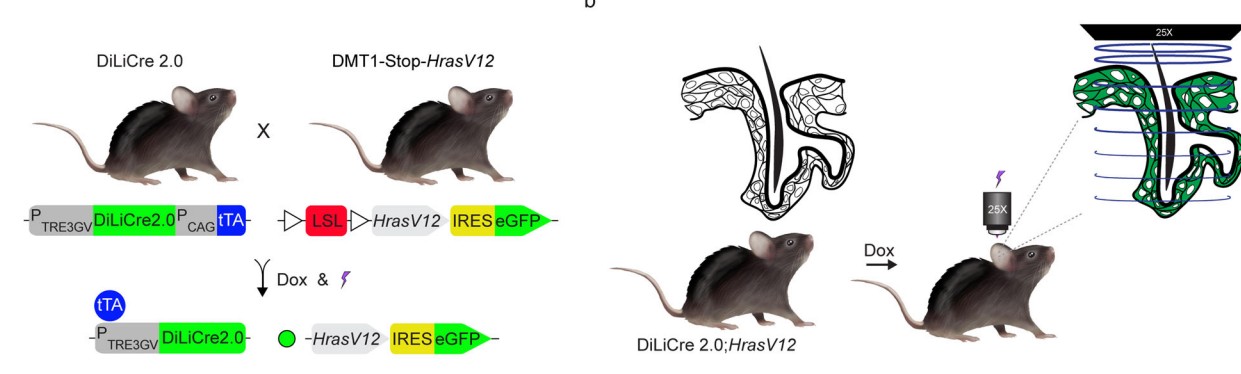

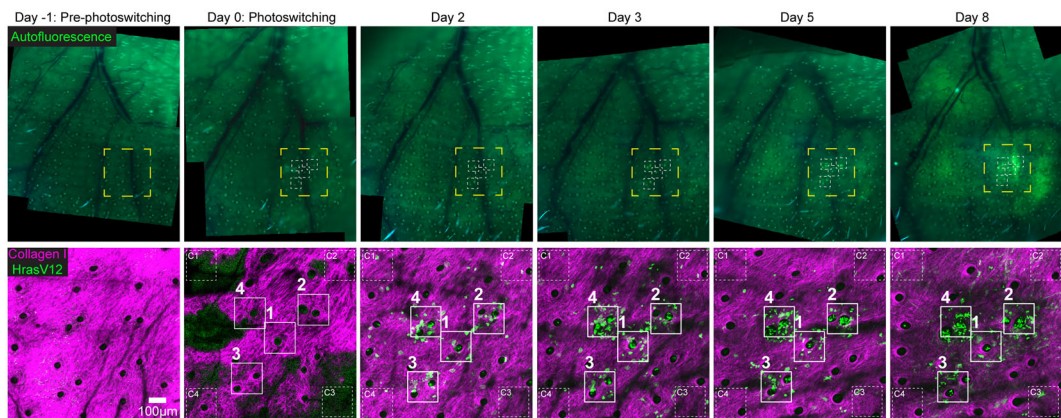

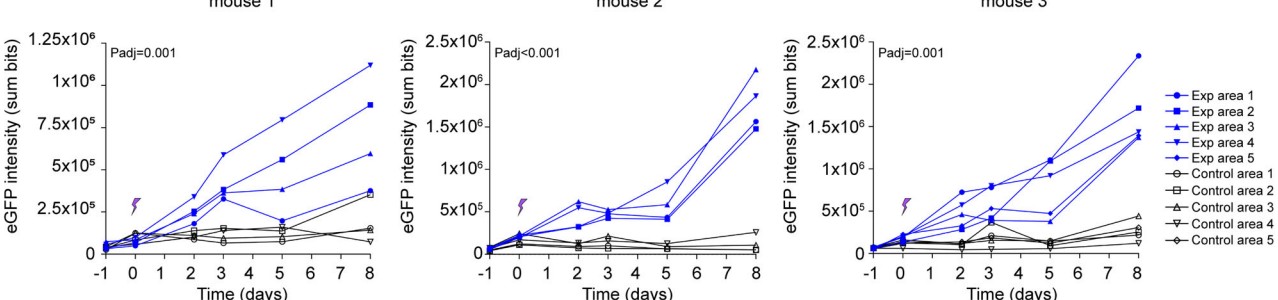

**Fig. 5 | DiLiCre2.0 mouse model is an efficient optogenetic tool to induce oncogenic cell transformations in vivo. a** Schematic representation of the breeding setup to generate the *DiLiCre2.0;HrasV12-eGFP* transgenic mouse model. LSL: Lox-Stop-Lox. **b** Schematic drawing showing the experimental setup for the photoactivation of the hair follicles. **c** Representative longitudinal images of the mouse ear skin upon DiLiCre2.0 photoactivation (HFs, mouse 1). The upper row shows widefield tissue overview images of the ear skin where the blood vessel pattern (black lines) and hair follicles (small white spots) can be identified. Within each image, the areas photoconverted are boxed using white dotted squares. The bottom row displays two-photon images corresponding to the yellow boxed areas.

Within these images, the photoactivated regions are annotated. $n = 3$ biologically independent experiments/mice were performed with similar results.
**d** Quantification of HrasV12-eGFP fluorescence over time in 405 nm light exposed and non-exposed regions. Control positions were selected randomly at least 150 μm away from the photoconverted areas. This experiment was performed in three mice and each graph shows the data of one individual. All biological replicates shown similar results. Between the two arms, the entire curves were compared statistically and the *P* values were determined by permutation test adapted from Elso et al. (2004). Source data are provided as a Source Data file.

EDTA and 2% of FBS in PBS. A broad FSC/SSC gate was followed by gates excluding doublets. Fluorescent positive cells were isolated using stringent gating adjusted using the corresponding control conditions (e.g. untreated cells).

For FACS quantification experiments to determine the percentage of recombination, cells were seeded (HEK293T: 8000 cells per well, C26: 2000 cells per well, s. intestine organoids: 5–10 small

intestine organoids per 50 μls-BME dropplet) on 24-well plates and treated with the corresponding doxycycline dosages 24 h later. Next day, media was refreshed and cells exposed to 405 nm light pulses (intensity and exposure time are indicated in the figure legends). Five to six days post-doxycycline treatment, cells were resuspended, dissociated (TripLe, Thermo Fisher Scientific, cat. no. 12604013) filtered using test tubes with cell strainer snap cap

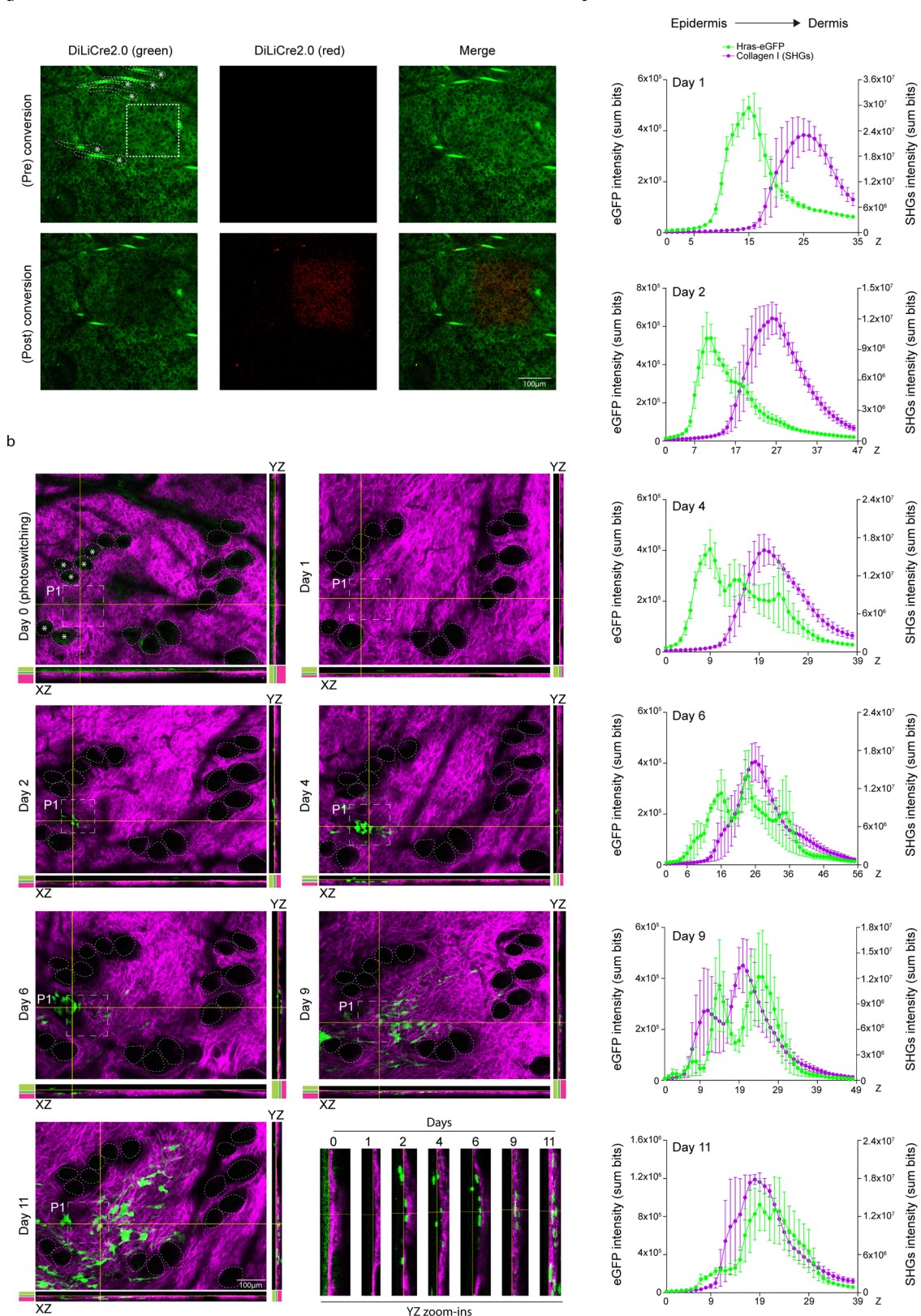

(Thermo Fisher Scientific, cat. no. 352235) and FACS buffer (2 % FBS and 2 mM EDTA in PBS), and analyzed by FACS LSRFortessa TM (BD Biosciences). For the Confetti reporter, an Aria Fusion BD sorter equipped with an additional 442 nm laser line was used. TO-PRO™−3 Iodide (642/661) was used as cell viability marker. For fluorescence detection, the following wavelengths/bandwidths were used, mTurquoise2: 470/20 nm; turboRFP: 610/20 nm with LP 600 nm. For confetti analysis, mCFP: 470/20 nm, nGFP: 510/20 nm with LP 495 nm; cYFP: 542/27 nm with LP 525 nm; cRFP: 610/20 nm with LP 600 nm. All the cells lines used in this study have been obtained from The Netherlands Cancer Institute cell line repository.

**Fig. 6 | Tracing the fate of cells that express HrasV12 upon localized DiLiCre2.0 activation. a** Photoconversion of DiLiCre2.0 at the basal layer region of the skin in *DiLiCre2.0;HrasV12-eGFP* animals. A Gaussian-blur filter was applied for visualization purposes. The photoconverted region is outlined by a white dotted square. Dotted lines and asterisks highlight the hair follicles of reference to help the reader localizing the same regions on the following panels. **b** Representative longitudinal two-photon images containing one photoconverted area (boxed by a white dotted square). A representative image of the z-stack and the XZ and YZ orthogonal views are annotated per imaging day. HFs are encircled by dotted lines to help the reader identifying the same structures along the experiment. Asterisks were used to make

reference to the same HFs annotated in **a**. Color codes denote the relative position of the epidermis (light green), the basal layer (dark green), and the dermis (magenta). *n* = 3 biologically independent experiments/mice were performed with similar results. **c** Quantification of HrasV12-eGFP and SHGs (days 1, 2, 4, 6, 9, 11) intensities along the z-axis from the in vivo experiments shown in panel **b**. Graphs show the location of HrasV12 recombinant cells within the skin which is determined by measuring the total HrasV12 (eGFP) and dermal collagen I (SHGs) intensities at each z-plane. Each line represents the average intensities of *n* = 5 photoconverted regions. One of those photoconverted region is shown in **b**. Data represents mean ± SEM. Source data are provided as a Source Data file.

## Nuclear-cytoplasm fractionation experiments

HEK293T cells engineered with DiLiCre1.0 or 2.0 were plated in 6-well plates ($3 \times 10^5$ cells/well). The day after, doxycycline was added to the corresponding wells (concentrations annotated in the corresponding figure legends). After 24 hours, cells were exposed to one single 405 nm light pulse (laser output power and exposure time are indicated in the corresponding figure legends). Nuclear-cytoplasm protein extracts were prepare following manufacturer instructions (ProteoExtract® Subcellular Proteome Extraction Kit, Merk). Nuclear-cytoplasm protein ratios were calculated to normalized the amount of protein loaded in the gels.

## Time-lapse confocal imaging experiments

Time-lapse confocal imaging experiments were performed using Leica TCS SP5 confocal microscope in 8-bit with 25X water objective (Fluotar VISIR 25.0 × 0.95) on 2D cultures (HEK293T and C26 cells) and 3D intestinal organoids derived from DiLiCre2.0;R26-Confetti mice. 2D cell lines were cultured in DMEM supplemented with 10 % fetal bovine serum (Thermo Fisher Scientific, 41966029) and 3D organoids were entrapped in 20 μls of a 1:3 mixture of BME (Cultrex, RGF BME type 2, cat. no. 3533-005-02) and basic organoid media [advanced DMEM/F12 medium (Thermo Fisher Scientific, cat. no. 12634-010) supplemented with GlutaMax 1X (Thermo Fisher Scientific, cat. no. 35050-068), Hepes 10 mM (Thermo Fisher Scientific, cat. no. 15630-056), and Pen/Strep 1X (Thermo Fisher Scientific, cat. no. 15140-122)]. 3D organoids were grown in organoid culture media [basic organoid media supplemented with Noggin-FC fusion 0.5 % (ImmunoPrecise, cat. no. N002), R-spondin1 10 % (conditional medium prepared in-house), B27 2 % (Thermo Fisher Scientific, cat. no. 17504-044), mEGF 50 ng/ml (Peprotech, cat. no. 315-09), N-acetylcysteine 1.25 mM (Sigma-Aldrich, cat. no. A9165)]. Experiments were carried on 8 well glass bottom μ-Slides (IBIDI, cat. no. 80827) seeding 2000 HEK293T cells, 16,000 C26 cells, and 20–50 small intestine organoids per well. The large number of C26 cells seeded is required in order to minimize the migratory capability of these cells and detect recombination. Time-lapse starts the day after seeding and stops after 48–72 hours depending on cell confluence. 5% CO2 supply was guaranteed by an Okolab gas chamber stage (Okolab, cat. no. H301).

For HEK293T and C26 cells the imaging settings had the following excitation (laser line) and emission wavelengths (HyD detectors). DiLiCre (green em.): ex. 488 nm and em. 498–540 nm; DiLiCre (red em.): ex. 561 nm and em. 577–700 nm; DiLiCre (photoconversion): ex. 405 nm (laser output power and exposure time are indicated on the figure legends); mTurquoise2: ex. 442 nm and em. 452–486 nm; turboRFP: ex. 561 nm and em. 577–700 nm. Line average 3 was applied on each sequential. Pre- and post- photoconversion images were acquired in the DiLiCre (green and red em.) channels. This was followed by image acquisition using the DiLiCre (green em.), mTurquoise2 and turboRFP channels every 15–30 min for 48–72 h.

For small intestine organoids the filter sets had the following excitation and emission wavelengths (HyD detectors). DiLiCre (green em.): ex. 488 nm and em. 495–545 nm; DiLiCre (red em.): ex. 561 nm and em. 571–661 nm; DiLiCre (photoconversion): ex. 405 nm (laser output power and exposure time are indicated on the corresponding

figure legends); mCFP: ex. 442 nm and em. 454–487 nm; nGFP: ex. 488 nm and em. 497–512 nm; cYFP: ex. 514 nm and em. 529-540 nm; cRFP: ex. 561 nm and em. 575–615 nm. Line accumulation 3 was applied on each sequential. Before/after photoconversion images were acquired in the PhoCl (green and red em.) channels. This was followed by image acquisition in the PhoCl (green em.), mCFP, nGFP, cYFP, cRFP channels every 90 minutes for 40–48 hours.

## Mice experiments

All animals and experiments were conducted under the guidelines by the Animal Ethics Committee of the Netherlands Cancer Institute and performed in accordance with institutional, national and European guidelines for Animal Care and Use. All the animal protocols and surgery and imaging procedures were reviewed and approved by the Animal Care Committee of the Netherlands Cancer Institute (codes: 9.2.9867, 9.2.9917).

To establish the DiLiCre2.0 knock-in mice, the TRE3GV promoter, the Cre-PhoCl-ERT2(x2) cassette, the Rb β-globin intron spacer and bGH polyA signal, the CAG promoter, the Teton3G transactivator sequence and bGH polyA signal were inserted into the F3-FRT targeting vector. The resulting plasmid DNA was lipofected together with a Flp-recombinase into ES cells from B6 (C57BL/6 J) mice containing the RMCE cassette (F3-puro-TK-FRT). Recombinant B6J-RMCE ES cells were selected with FIAU (a negative selection against HSV-TK (herpes simplex virus thymidine kinase) to kill the non-recombinant cells that still retain the RMCE cassette). Recombinant alleles were verified by Southern Blot to confirm single copy integration. ES cells from single clones (B8 and B10) were each injected into blastocists of B6 (C57BL/6N) mice for posterior PCR determination of ES contribution (>90 %) in the chimeric progeny. Control and doxycycline (100 μls of PBS containing 1 mg of doxycycline by IP injection) treated F2 generation animals (8–12 weeks old) were sacrificed and organs were collected in PLP buffer (periodate-lysine-paraformaldehyde) to characterize the expression of DiLiCre2.0 using a slide scanner Zeiss Axioscan and Leica TCS SP5 confocal microscope.

To evaluate the recombination efficiency of DiLiCre2.0 system in these mice, DiLiCre2.0 transgenic mice were crossed with B6 R26-Confetti mice (a gift from Hans Clevers lab). Recombination was measured ex vivo by isolation of small intestinal crypts (see time-lapse confocal imaging experiments section) or directly by intravital imaging through the implantation of an abdominal window[6]. In brief, anesthetized DiLiCre2.0;R16-Confetti mice treated with Rimadyl (25 μg/ml in drinking water) and Temgesic (0.003 mg/25 g of BW, i.p.) were submitted to skin and abdominal wall incision through the linea alba to have access the abdominal cavity. PBS moisturized cecum and small intestine were placed on top of the abdominal-side of the imaging window using sterile cotton swabs. Extra adhesion was performed by covering the mesentery tissue with few drops of CyGel (Abcam) and intestinal wall with histoacryl glue (Braun). Finally, imaging window was fixed by a 5-loop suture to the skin and the abdominal wall. In vivo expression of DiLiCre2.0 was induced by intraperitoneal injection of doxycycline (100 μg) in PBS (100 μls) per 25 g of body weight. For multiphoton imaging, mice were placed in a custom-designed imaging box on the microscope stage and were kept under constant

anesthesia with imaging box and microscope adjusted to 34.5 °C using a climate chamber. Imaging was performed on an inverted Leica SP8 Dive system (Leica Microsystems, Mannheim, Germany) equipped with four tunable hybrid detectors. All images were collected using LAS X software at 12 bit and acquired with a 25x water immersion objective with a free working distance of 2.40 mm (HC FLUOTAR L 25x/0.95 W VISIR 0.17). Confetti fluorophores were excited and detected as follows: mCFP (850 nm, HYD1: 420–430 nm for SHGs and HYD2: 450–484 nm), nGFP (960 nm, HYD3: 495–518 nm), cYFP (960 nm, HYD4: 538–560 nm), and cRFP (1040 nm, HYD3: 561/605 nm). For multi-day repetitive imaging, an overview scan of the visible tissue area was acquired. After this, defined areas of interest were imaged by taking z-stacks with 3 μm step size. The same imaging fields were retraced during subsequent imaging sessions using the imaging overviews of the first imaging session. Photo-switching was performed within that areas at two different locations. These zoom-in areas containing ~1–4 crypts were exposed to 405 nm laser at 5% intensity (imaged with 7.95 ± 2.19 mW laser output power for 45 s. To revisit the same regions over time, vasculature was used as landmarks.

To induce cell transformation in vivo, DiLiCre2.0 mice were crossed with *DNMT1-CAG-loxP-STOP-loxP-HrasV12-IRES-eGFP* knock-in mice (kindly provided by Yasuyuki Fujita's Lab)[27]. The new transgenic mice do not express the oncogene *HrasV12* unless Cre is activated in cells, which will induce the excision of the transcription roadblock and allow the simultaneous expression of the oncogene and the fluorescence protein eGFP. Thus, eGFP can be used as a reliable readout of the oncogene expression in vivo. In vivo expression of DiLiCre2.0 was induced by intraperitoneal injection of doxycycline (100 μg) in PBS (100 μls) per 25 g of body weight. Imaging was performed as previously described for the intestine. Photoconversion was achieved by 405 nm laser at 405 nm laser at 3% intensity (imaged with 5 ± 1.34 mW laser output power for 60 s. For testing light-induced *HrasV12* recombination, 4–5 follicular regions of interest were selected for photoconversion per mouse. For positional cell-tracing experiments of HrasV12 cells, 3–5 interfollicular regions were selected for photoconversion per mouse. DiLiCre2.0 (green em.) was collected on HYD 3 (505-525 nm) upon 960 nm excitation. DiLiCre2.0 (red em.) was collected on HYD 4 (580-590 nm) upon 1040 nm excitation. HrasV12-eGFP was detected by excitation at 960 nm and emission collected on HYD3 (505-525 nm). SHGs was detected on HYD2 (475–485 nm). Extra light was collected in the 574–605 nm range for image subtraction. To revisit the same regions over time, organizational clusters of hair follicles and vasculature were used as landmarks. Eight-to-ten old week mice were used.

All animal imaging experiments were performed under anesthesia and supplying extra warmed saline solution to avoid dehydration. Heterozygous mice for each transgene were used in all experiments. For PCR genotyping of mice, *DiLiCre2.0*F: TGCTCGCACGTACTTCATTCC, *DiLiCre2.0* R: GACAGAAGCATTTTCCAGGTATGC, *R-26-Confetti*F: GCTGAACTTGTGGCCGTTTA, *R-26-Confetti*R: GCGCATGAACTCTTTGATGA, *HrasV12*F: CACTGTGGAATCTCGGCAGG, *HrasV12*R: CGGCAATATGGTGGAAAATAACA.

### Imaging processing and analysis

All the imaging data shown in this study were processed in Fiji software (National Institute of Health)[28], using macro scripting and machine-learning algorithms (WEKA segmentation[29]). For visualization purposes specific filters (e.g. Gaussian blur) were used (annotated in the figure legends when applied), but all quantifications were performed using raw images. All the analytical workflows are described in the corresponding figures captions. In brief, data analyses of fluorescence intensities in time-lapse confocal experiments were performed upon subtraction of background signal as measured in areas without cells. Signal was measured in regions of interest: either the area that is covered by cells or the area that is covered by RFP positive nuclei. To obtain these regions of interest, bright field (BF) or merged BF-RFP

confocal images were used for segmentation. Segmentation is based on pixel classification (Fiji WEKA plugin) and was performed as follows. (i) The training set of ~5 images is annotated, indicating the features of interest (cell covering area or RFP positive nuclei), cell debris and well plate background. Weka default settings were applied. (ii) The classifier is retrained until it performs adequately using default training features. (iii) After ensuring appropriate performance, the classifier is saved (Supplementary software). (iv) 6 to 8 z-stacks are acquired per experimental condition. (v) Images are loaded into Fiji and (vi) the classifier created in i–iii is applied. The images are converted to binary and particles analysis applied to measure the total area covered by cells or the nuclear RFP intensity. Finally, total intensities are divided by the corresponding areas to estimate cell recombination at each time point (Supplementary Fig. 3f, g).

To quantify fluorescence intensities in time-lapse confocal experiments in small intestine organoids, we performed subtraction of background signal measured from regions without cells. Then, a channel leak-through of nGFP into the cYFP channel was determined by dividing the cYFP by the nGFP background-subtracted values in the reference region. The cYFP channel was then corrected by the formula $cYFP_{bs}-(nGFP_{bs}*LF)$ with bs being the background-subtracted channels and *LF* the leak-through factor. Pixel-classification (see above) was performed on each fluorescence channel individually to get the binary masks, after which particle analysis was run. Particle size (s (μm²)) and circularity (cir) were adjusted according to the spatial diversity of fluorophore expression in the Confetti cassette (mCFP-s: 45–1000, mCFP-cir: 0.4–1; nGFP-s: 20–70, nGFP-cir: 0.5–1; cYFP/cRFP-s: 35–1000, and cYFP/cRFP-cir: 0.3–1). Pixel-classification was also applied on bright field images to determine the organoid size. Total C/G/Y/RFP intensities were calculated and divided by the organoid size to estimate cell recombination at each time point (Fig. 3d).

To quantify fluorescence intensities in vivo in the small intestine, correction for XYZ drifts was performed by XYZ manual adjustments. This was followed by tile stitching using Fiji's BigStitcher plugin[30] (specifically: a two-round phase correlation with downsampling 4 and 2, respectively, and global optimization). Then, we performed background subtraction and leak-through correction was performed as described above. Particle size (s (μm²)) and circularity (cir) were adjusted accordingly to the different locations of the fluorophore expression in the Confetti cassette (mCFP-s: 100–5000, mCFP-cir: 0.4–1; nGFP-s: 70–300, nGFP-cir: 0.5–1; cYFP/cRFP-s: 100–5000, cYFP/cRFP-cir: 0.3–1). At each time point, total C/G/Y and RFP intensity values were calculated on each z-plane, subsequently summed and divided by the number z-steps to estimate the rates of cell recombination (Fig. 4e, and Supplementary Figs. 7c–e and 8d).

Finally, in skin analysis, tile stitching (Leica LAS X stitching 3D module) and correction for XY drifts (Fiji BigWarp plugin[31] and manual correction by landmark annotation) were performed. Then, a channel leak-through of DiLiCre2.0/HrasV12 (green) into the red ("empty channel") channel was determined in a similar way as described above. For testing light-induced HrasV12 recombination, total eGFP fluorescence was measured in each ROI (*n* = 4–5) and data divided by the number of z-steps to estimate cell recombination at each time point (Fig. 5d, and Supplementary Figs. 9c and 10c). For positional cell-tracing experiments of HrasV12 cells, the total intensities of HrasV12-eGFP and collagen I (SHGs) were measured at each z-plane in all the interfollicular regions photoconverted. These values were averaged per mouse and plotted (Fig. 6c, and Supplementary Figs. 9c and 11a, b).

### Statistical analysis

Before *P* values calculations, data was tested for normality (K-S or Shapiro Wilk). Upon normality, two-tailed *t*-tests were run. In multiple testing analysis, Bonferroni method was applied to post hoc multiple comparisons after one-way anova. Upon non-normality, the non-parametric test of choice was Mann−Whitney (if *n* ≥ 3) (for FACS data

comparisons). P values were calculated using SPSS (IBM SPSS Statistics for Windows). To compared recombination rates over time in time-lapse imaging experiments, an adaptation of the permutation test used by Elso et al., (2004)[32] was applied ($n = 1000$ permutations). In brief, the analysis is based on the F-score from an anova test, were we compare the original score to those obtained by permutation, using an empirical cumulative distribution to get the P values.

## Statistics and reproducibility
No statistical method was used to predetermine sample size due to the nature of the study. Only one data point was excluded in Supplementary Fig. 5b. This data point shown abnormal high number of dead cells when analyzed by flow cytometry. All the main experiments in this article were repeated at least three times independently. In addition, results were validated using orthogonal methods such as time-lapse confocal microscopy and flow cytometry in parallel. All the in-vivo experiments were performed including internal and external control conditions. Randomization was applied when selecting the positions to be exposed to the 405 nm laser light. Other than that, no randomization was applied due to the nature of this study. Blinding was applied when collecting the organs for testing the expression of DiLiCre2.0 in the murine tissues. Other than that, no blinding was applied due to the nature of this study.

## Reporting summary
Further information on research design is available in the Nature Research Reporting Summary linked to this article.

## Data availability
Source data are provided with this paper. Software files (Weka segmentation files) are provided with this paper as Supplementary Software. Source data are provided with this paper.

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

## Acknowledgements

The authors would like to thank the van Rheenen group members for critically reading this manuscript. This work was supported by the

European Research Council (consolidator grant 648804 to J.v.R.), ZonMW of the Nederlandse Organisatie voor Wetenschappelijk Onderzoek (NWO)(VICI 09150182110004 to J.v.R.), Doctor Josef Steiner Foundation (to J.v.R.), and the fellowship from EMBO ALTF 437-2017 (M.V.).

## Author contributions

M.V. and J.v.R. conceived the study. M.V. performed the in vitro experiments and intravital microscopy experiments. M.V. and C.E.J.P. generated the DiLiCre2.0 mESCs and mouse models. M.V. and L.B. tested the alternative pa-Cre systems. M.V. performed all the analysis and B.v.d.B. helped with analyzing and interpreting the imaging data. M.V. and K.J. worked together to design the LED custom chambers and optimize the light exposure conditions. M.V. wrote the first draft and generated all the figures. All authors contributed to writing and have approved the manuscript.

## Competing interests

The authors declare no competing interests.
