## [Peer Review File · Nature Communications]

Reviewers' Comments:

Reviewer #1:

Remarks to the Author:

Vizoso and their colleagues reported the development of DiLiCre2.0, a newly engineered Cre recombinase, which is designed to be controlled by doxycycline and blue light. The strengths of this study are 1) applying DiLiCre2.0 in small intestine-derived organoid model; 2) generating DiLiCre 2.0 mouse line; 3) applying DiLiCre2.0 for intestinal study with Confetti fluorescent reporter mouse line; 4) applying DiLiCre2.0 for hair follicle study with HraV12 overexpression. The results suggest that DiLiCre 2.0 seems to be efficient and promising for further application in vitro and in vivo using human cellular models and mouse models, respectively. However, this reviewer obtained several concerns. It'd be great if the author could address the below concerns to improve their manuscript.

Major concerns:

- 1) While DiLiCre 2.0 seems to be efficient according to their results, it'd be better for the authors to conduct several benchmark experiments comparing to the other photo-controllable Cre such as PA-Cre series (ex. using the same illumination, cells and reporters).
- 2) Can two-photon illumination (ex. 920nm) activate DiLiCre2.0 in vivo?
- 3) Is it possible that endogenous estrogen induces Cre-lox recombination in vivo in DiLiCre2.0 mouse because spontaneous Cre-ERT2 activity is a major concern in the field (<https://pubmed.ncbi.nlm.nih.gov/28409408/>)? Why don't the authors use NES (nuclear export signal) instead of ERT2 although this ERT2/estrogen-related concern is common in the field?
- 4) The author mentioned, "Although a few confetti labelled cells also appeared in the doxycycline treated and non-light exposed organoids, light... (Page 8, Line 186)" regarding Fig. 3d,e. However, in the experiments shown in Figures 4 and 5, "doxycycline-treated and non-light exposed" mice were not examined as an essential negative control for their study in vivo. Their selected control positions can be possibly biased still. It'd be better if the authors could provide such negative control results using DiLiCre2.0; Confetti mouse intestines and hair follicles in vivo with doxycycline treatment and dark condition. If various fluorescent patterns are observed in the tissues, DiLiCre2.0 system seems to be leaky, no longer light-controllable and useless for further in vivo applications.

Minor concerns:

Fig. 1b: it is difficult to see the words inside cells because too small.

Fig. 1c: is there any reason why stronger light was required for C26 cells compared to HEK293T

Fig. 1d: it'd be better if the authors could provide quantification of nRFP+ cells to demonstrate the efficiency of DiLiCre 1.0.

Fig. 2d: it'd be better if the authors added the raw flowcytometry results as they did for Fig. 3b.

Fig. 3b: the flowcytometry images are too small. It'd be better to move the whole images to supplementary material using larger images with better resolution.

Fig. 5c,d It'd be better to analyze the number of GFP+ cells as well as GFP intensity because other area seem to have GFP+ cells.

Reviewer #2:

Remarks to the Author:

Cre recombinase is a valuable tool for making genetic modifications in live animals. Several light activated Cre recombinases have been reported. These systems could potentially be used for induction of recombination events in specific tissues and at specific times, which could enable new studies of physiology, development, cancer, etc. However, previous light activated Cre systems exhibit substantial leakiness in the dark, which may have prevented their deployment in vivo.

The authors address this challenge by engineering a lentiviral expression system wherein a recent violet-light activated engineered Cre protein is expressed from a stringent doxycycline inducible promoter. They further increase Cre activity while reducing leakiness by altering the engineered domain architecture of the photactivatable Cre protein in several ways. The performance is improved relative to previous systems, though there is still leakiness, especially in vitro. However, the authors find that the system enables fairly strong optical activation with relatively low impact of leakiness in vivo settings including lineage tracing and oncogenesis. By inducing an oncogenic gene, they make some interesting observations of potential cell migratory processes in cancer.

Overall, I think this is an interesting and impactful study. Other groups should find this tool useful for in vivo studies. I have some concerns with the manuscript, which are detailed below. If this concerns can be addressed, I recommend publication in Nature Communications.

Concerns

It is not clear to me how PhoCl photoactivation causes Cre recombinase to become dissociated from the ERT domains so that it can translocate to the nucleus. The authors should elaborate on the mechanism of this process.

The authors claim that the green-to-red switch of PhoCl enables "optimization of light exposure, rendering it ideal for our experiments" (line, 92). I do not understand what the authors mean by this statement. What about light exposure needs to be optimized? What does optimal mean in this context? How does PhoCl enable this optimization, compared to other photoswitchable domains used to control Cre?

The authors show that DiLiCre0.0 is photosilent, and go on to fix it by removing the N-terminal ERT2-PhoCl domain, which is nice. However, they provide no explanation for why this domain was contributing to photosilencing, nor why they chose to remove it as opposed to the C-terminal domain. The authors should at least present a hypothesis for why version 0.0 does not work and how removal of this domain restores function.

The timelapse data in Figure 1d is not clearly presented. There are numbers with increasing values overlaid on images of cells. However, those numbers have no units, and are not defined in the figure legend. The presentation should be clearer.

Also, the response of the cells seems much stronger in panel c than in panel d. The authors state that panel d is a timelapse series that led to the data in panel c. Why is the response so much weaker in panel d? Any differences between these two experiments should be clarified.

I don't understand the statement on lines 126-128. Shouldn't removal of an ERT2 domain result in a decrease in cytoplasmic sequestration of Cre?

The authors do not define how the cytometry data are plotted in the figure legends (e.g. Extended Data Figure 2f). What are the multiple data points in each data set?

The claim that removal of one of the two ERT2 domains from DiLiCre0.0 leads to a significant amount of background activity (lines 139-140) is not supported by the data provided (Extended Data Fig. 2f). The authors show no data comparing DiLiCre0.0 to DiLiCre1.0, they only compare different expression levels of the latter.

Figure 3b is not readable. There is far too much information shown. The feature and text sizes are extremely small. The presentation needs to be reworked. All of the raw data that is currently being shown could be presented in the Extended Data.

How is it that the fold change of recombination (Figure 2c) is so much higher in *in vivo* (~20-fold; Figure 3c) than *in vitro* (~2-fold; Figure 2d)

I do not understand the treatment conditions in Figures 3d and e. For example, what do the 3 columns in Figure 3d represent? Different times of light exposure? If that's the case, then what is meant by "light control" in row 1? Does that actually mean no dox was added? The presentation of conditions in these figure panels should be clarified.

Reviewer #3:

Brief summary of the research

Regulation of gene expressions in a spatiotemporal manner is required to understand the role of the specific gene product and related biological mechanisms. However, most chemical inducers are toxic to animals and CRISPR-based systems are hard to deliver guide RNA to a target organ. To detour the limitations of existing genome inducer tools, the authors improved the light-inducible Cre recombinase system.

The authors introduced a doxycycline-inducible TETon promoter to the existing PhoCI system to create the two-lock photoactivatable Cre recombinase (DiLiCre1.0). Further fusion of the additional ERT2 domain reduced the background recombination without a loss of efficiency (DiLiCre2.0). DiLiCre2.0 is activated by short light exposure times with an optimized ratio of light-induced and background Cre activity. Additionally, DiLiCre2.0 was characterized in cancer cells and organoid systems.

Finally, *in vivo* induction of mutagenesis and following cell-tracing is possible using DiLiCre2.0. Authors induced spatiotemporal HrasV12 mutagenesis in basal layer cells. They observed that the HrasV12 mutation had cells migrate toward the collagen dermal layer, which is the opposite behavior compared to normal cells of the basal layer. Such positional tracing of skin cells showed the

applicability of DiLiCre2.0 to *in vivo* cell-tracing which enables spatiotemporal observation of biological phenomena.

Pros.

- 1) Introduction of doxycycline-inducible TETon system into developed photoactivatable Cre improves limitations of existing light-inducible Cre recombinases by reducing background activity.
- 2) Additional control of light-inducible Cre enables spatiotemporal regulation of Cre recombinase and following recombination *in vivo* in the skin and intestinal crypt.
- 3) Positional cell tracing of a basal and collagen layer with DiLiCre2.0 knock-in mice suggests the applicability of the light-induced Cre system for lineage tracing of mutated cells in the skin.

Cons.

- 1) It is hard to estimate the relative efficiency and background activity of DiLiCre2.0 due to the lack of comparison of DiLiCre2.0 with other developed photoactivatable Cre systems.

1. In **Extended Data Fig.1j**, pCMV-Cre induced a conversion from membranous mTq2 signal into nuclear turboRFP signal. There was an obvious decrease in mTq2 signal while DiLiCre1.0 failed to diminish the mTq2 signal after light-activation in **Figure.1d**. We recommend measuring the intensity change of membranous blue signal (pre-conversion versus post-conversion) to fully support that the RFP signal emerged by Cre recombinase and following recombination. There should be a mTq2 intensity difference even if such difference was generated by a difference of efficiency between pCMV-Cre and DiLiCre1.0.

2. In **Figure.2c and 2d**, DiLiCre2.0 showed better performance than DiLiCre1.0 for Cre activity. However, I questioned the relative efficiency and background Cre activity compared to other existing photoactivatable Cre recombinases, such as paCre and PhoCl, as you mentioned. It would be more powerful to display the performance of DiLiCre2.0 with other developed photoactivatable Cre recombinase tools to argue that DiLiCre2.0 well-overcame presenting limitations.

3. In **Figure.4**, the authors have emphasized the spatiotemporal control of DiLiCre2.0 activity *in vivo*. However, spatial-specific Cre activation is not mentioned in the article. Though **Figure.4** legend indicates the spatial control with the word 'ROI', it would be better to point out the spatial-specific photoactivation in the main paragraph.

4. In **Figure.5c**, the authors have selected four photoactivated regions and four control regions but there are five samples in the graph in **Figure.5d**. Also, the authors have mentioned in the **Method** that the GFP signal was measured in 4-5 ROIs. Is missing ROI in the other window? Additionally, I questioned whether delicate spatiotemporal control is possible with DiLiCre2.0 *in vivo* by activating the region 1-4 sequentially and inducing the recombination in different time window.

5. How can authors distinguish green signals between DiLiCre2.0 and HrasV12 other than time point in **figure.5**? Does a line 228 ("the amount of fluorescence of HraV12 is on order of magnitude larger than the fluorescence of DiLiCre2.0") explain the distinguishable property of DiLiCre2.0 and HrasV12 signals?

6. Was there any experimental try to express and activate the DiLiCre2.0 other than skin? I questioned any application possibility of DiLiCre2.0 in other organs deeper than skin and intestinal

crypts.

Reviewer #1:

Vizoso and their colleagues reported the development of DiLiCre2.0, a newly engineered Cre recombinase, which is designed to be controlled by doxycycline and blue light. The strengths of this study are 1) applying DiLiCre2.0 in small intestine-derived organoid model; 2) generating DiLiCre 2.0 mouse line; 3) applying DiLiCre2.0 for intestinal study with Confetti fluorescent reporter mouse line; 4) applying DiLiCre2.0 for hair follicle study with HraV12 overexpression. The results suggest that DiLiCre 2.0 seems to be efficient and promising for further application in vitro and in vivo using human cellular models and mouse models, respectively. However, this reviewer obtained several concerns. It'd be great if the author could address the below concerns to improve their manuscript.

Reply:

We would like to thank this reviewer for his/her time to review our manuscript and appreciate the constructive feedback. As we will detail below, we have addressed all points, and feel that this helped us to construct a much improved manuscript.

Major concerns:

1) While DiLiCre 2.0 seems to be efficient according to their results, it'd be better for the authors to conduct several benchmark experiments comparing to the other photo-controllable Cre such as PA-Cre series (ex. using the same illumination, cells and reporters).

Reply:

There are two different photoactivatable Cre (PA-Cre) systems: 1) based on dimerization (e.g. Magnet, split-Cre), and 2) based on cleavage (PhoCI). Our DiLiCre system belongs to the second category. The dimerization series is activated by blue light (460-480nm), whilst the PhoCI series is activated by violet light (~405nm). Therefore, a direct comparison between those series using the same illumination and reporters is not possible. Instead, we investigated the various characteristics that are required for successful use of these PA-Cre series for mouse experiments.

Comparison to dimerization systems:

Although different activation wavelengths between dimerization based photoactivatable Cre systems and our DiLiCre2.0 system renders a side-by-side comparison difficult, for the revised manuscript we have compared some basic traits. We tested two different split-Cre systems: the Magnet system by the Sato group (Kawano et al, 2016) and the PA-cre2.0 by the Tucker group (Meador et al., 2019). Both are based on a split-Cre strategy, where the Cre subunits are each fused to blue-light sensitive dimerization domains that reconstitute Cre enzymatic activity. The Magnet system published by the Sato group (Kawano et al, 2016) is based on fungal dimerizing proteins, while the PA-Cre2.0 by the Tucker lab (Meador et al, 2019) is based on Arabidopsis proteins. Both were reported to have a signal to noise ratios >200 (defined as the ratio between Cre activity in the presence of 460-480nm blue light versus activity in the dark).

We first tested the Magnet system, as it was the first to be published and was reported to have excellent signal/noise figures. However, Duplus-Bottin and colleagues (2021) recently published that the Magnet system in their hands led to much less favorable ratio of photo-induced Cre-activity and background Cre activity than reported in the original report. Indeed, our transfection experiments suggested higher levels of background recombination than those reported by the original manuscript (**Extended data Fig. 12a** of the revised manuscript). We then speculated that constitutive expression could be responsible for the elevated background noise, so we tested whether an inducible construct would solve this problem (**Extended data Fig. 12b** of the revised manuscript). In our setting however, the overall recombination efficiency was far lower than the published results. Upon doxycycline treatment (100nM), background activation in the absence of light was unacceptably high, up to 35% of the total signal (**Extended data Fig. 12b** of the revised manuscript). We established that activation time was not the issue, as blue light

irradiation times between 2 min and 20min minimally affected Magnet efficiency (**Extended data Fig. 12b** of the revised manuscript). Therefore, our observations are in line with other recent reports (Duplus-Bottin et al., 2021) that the signal-to-noise ratio of the Magnet is too low for a practical tool.

We then tested the PA-Cre2.0 system published by the Tucker lab in 2019 and followed a similar strategy. A side-by-side comparison with the Magnet system showed that this construct had an intrinsic lower noise (**Extended data Fig. 12a** versus **c** of the revised manuscript). Although *in vitro* PA-Cre2.0 Tucker had better overall efficiency and reduced noise compared to the Magnet system, this system showed some ground problems preventing *in vitro* and *in vivo* use: we were unable to stably integrate or avoid the silencing of the PA-Cre2.0 in HEK293T cells as it was found by our multiple rounds of FACS enrichment for PA-Cre2.0 (**Extended data Fig. 12d** of the revised manuscript). More than 90% of the mCherry-PA-Cre2.0 positive cells selected in first round were lost upon few cell passages. We reported this observation to Tucker's lab and they confirmed that they experienced the same problems and their unsuccessful attempts to get stable integrations.

From those experiments, we concluded that the published dimerization-based photoactivatable Cre are not usable for *in vivo* use because 1) too much background, 2) too low recombination efficiency, 3) cannot generate stable lines (due to the lack of permanent integration or silencing).

Lastly, and not unimportantly, the blue activation light possess a large limitation for the *in vivo* use of dimerization systems. Most reporter models that are currently available are based on GFP-like fluorescence that require blue excitation light. This blue excitation light also activated the dimerization systems rendering it unusable for Cre lineage tracing experiments.

In the revised manuscript, we describe the limitations of these dimerization systems at page 10 and lines 265-269.

Comparison to photocleavable (PhoCI) based systems:

We also compared our DiLiCre2.0 with the PhoCI series. We have tested three modified versions of the original photocleavable Cre system containing the 405nm-photoswitchable protein PhoCI (Zhang et al., 2017). Similar to our strategy when testing the dimerizing systems (Magnet and PA-Cre2.0), all our adapted versions of the photocleavable Cre system worked under TetON response elements and doxycycline induction in order to minimize the background recombination levels.

First, our data shown that DiLiCre0.0, a doxycycline inducible version of the original construct used by Zhang and colleagues 2017, failed to induce efficient recombination of cells both in the absence of presence of light (**Extended Data Figure 2a-i** of the revised manuscript). However, DiLiCre1.0, which contains only one PhoCI-ERT2 tandem repeat, shown higher background recombination compared to DiLiCre0.0 (**Extended Data Figure 3a, e** and **i** of the revised manuscript) and most importantly, light-induced Cre recombinase activity (**Figure 1d** and **Extended Data Figure 3a, g** and **j** of the revised manuscript). The induction of DiLiCre1.0 with doxycycline at 100nM in HEK293T cells, revealed $8.04 \pm 1.99\%$ of background recombination and $25.44 \pm 3.60\%$ of light-induced Cre recombinase activity (**Extended Data Figure 3j** of the revised manuscript). In our attempt to improve the ratio of light-induced and background activity of the Cre recombinase, we developed DiLiCre2.0, which contains two ERT tandem repeats. This modification led to an increase in induction efficiency and a lower background activity compared to DiLiCre1.0 (**Figure 2a-to-d** of the revised manuscript).

Secondly, we were able to achieve stable integrations of all our DiLiCre developed constructs *in vitro* and ultimately DiLiCre2.0 *in vivo*. This represents an important advantage respect the PA-Cre Tucker construct series based on Arabidopsis light responding proteins.

Conclusion:

From all these experimental comparisons, we concluded that DiLiCre2.0 was the molecular tool of choice for lineage tracing experiments and we therefore have developed the new photoactivatable Cre transgenic mouse model. Indeed, our conclusion is confirmed by our successful *in vivo* experiments with this mouse model (**Figures 4-to-6**).

2) Can two-photon illumination (ex. 920nm) activate DiLiCre2.0 in vivo?

Reply:

We agree that this is an important question, and for the revised manuscript, we have performed multiple experiments to test whether two-photon illumination is able to activate DiLiCre2.0. We first would like to stress that for the activation of DiLiCre2.0, the PhoCl unit needs to be photoconverted (green-to-red) so that the ERT2 unit breaks off, which enables the Cre unit to translocate to the nucleus (**Fig 1a** and **b**, and **Extended Data Fig. 1** of the revised manuscript). The photoconversion of the PhoCl unit of DiLiCre2.0 happens efficiently with violet light (~405 nm), so for two-photon illumination, it is expected to be efficient in the range of 805-820nm.

To test this, we exposed organoids lines expressing DiLiCre2.0 to various infrared wavelengths (805-820)(see **Extended Data Fig. 13** of the revised manuscript). We observed that the PhoCl unit of DiLiCre2.0 is efficiently photoconverted by 405nm light, but not by two-photon illumination (**Extended Data Fig. 13** of the revised manuscript). Moreover, we observed that increased laser power induced photobleaching but not photoconversion. Importantly, the inability to be photoconverted by two-photon illumination is shared by other photoactivable proteins, such as Dendra2 (Dempsey et al., 2015 Nature Methods; Gurskaya et al., 2006 Nature Biotechnology). In the revised manuscript, we show this data in **Extended Data Figure 13** and mention this observation at page 12 and lines 294-297.

3) Is it possible that endogenous estrogen induces Cre-lox recombination in vivo in DiLiCre2.0 mouse because spontaneous Cre-ERT2 activity is a major concern in the field (<https://pubmed.ncbi.nlm.nih.gov/28409408/>)? Why don't the authors use NES (nuclear export signal) instead of ERT2 although this ERT2/estrogen-related concern is common in the field?

Reply

During the development of DiLiCre2.0 we also considered the possibility to use a NES unit instead of a ERT2 unit. As suggested, we had also engineered a NES-DiLiCre construct (based on the same sequence published by Zhang et al., 2016 and Meador et al., 2019) and transduced it into a cell line containing the nRFP Cre reporter (**Rebuttal Figure 1a-to-c**, see below). Unfortunately, our flow cytometry data indicated that this construct failed to provide a good signal-to-background recombination ratio (**Rebuttal Figure 1a-to-c**, see below). For that reason, we decided to further develop the DiLiCre system based on the ERT2 domains, which have been successfully used in other non-optogenetic CreERT2 mouse models by a vast number of labs (including our of lab).

a

b

c

Rebuttal Figure 1. Evaluation of the performance of DiLiCre with a nuclear export signal.

a, Schematic representation of the NES-DiLiCre photoactivatable Cre recombinase system containing the tetracycline response element promoter, the nuclear export signal (adopted from Zhang et al., 2016 and Meador et al., 2019), the PhoCl fluorophore, and the Cre recombinase. Moreover, the floxed nuclear-RFP cassette is shown. Arrowheads indicate the light-breaking points of the PhoCl fluorophore.

b, Evaluation of the recombination levels in HEK293T cells stably expressing the NES-DiLiCre system and the Cre reporter nuclear-RFP. Four different conditions were tested: untreated cells, cells exposed to doxycycline or light, and combined treatment (100nM or 1000nM dox and 0.5mW/mm² 90s-light pulse). FACS plots are shown.

c, Quantification of the nuclear-RFP excision as a measurement of Cre recombination among the different conditions depicted in panel b.

4) The author mentioned, “Although a few confetti labelled cells also appeared in the doxycycline treated and non-light exposed organoids, light... (Page 8, Line 186)” regarding Fig. 3d,e. However, in the experiments shown in Figures 4 and 5, “doxycycline-treated and non-light exposed” mice were not examined as an essential negative control for their study in vivo. Their selected control positions can be possibly biased still. It’d be better if the authors could provide such negative control results using DiLiCre2.0; Confetti mouse intestines and hair follicles in vivo with doxycycline treatment and dark condition. If various fluorescent patterns are observed in the tissues, DiLiCre2.0 system seems to be leaky, no longer light-controllable and useless for further in vivo applications.

Reply:

We agree and have added these controls in **Extended Data Fig. 8** (intestine) and **Extended Data Fig. 10** and **Video 5** (skin) of the revised manuscript. These negative controls confirm that the background recombination levels in doxycycline-treated and non-light exposed animals, are in accordance with the background recombination levels measured in the experimental mice (at regions not exposed to the 405 nm laser light).

Minor concerns:

Fig. 1b: it is difficult to see the words inside cells because too small.

Reply:

Thanks, we have made them bigger now.

Fig. 1c: is there any reason why stronger light was required for C26 cells compared to HEK293T

Reply:

HEK293T cells are well known for high expression levels of exogenous transfected constructs. Indeed, the expression level of DiLiCre2.0 is higher in HEK293T cells than in C26 cells. It is important to stress that the DiLiCre system performs better with higher light intensities or longer exposure times. Therefore, in order to achieve similar levels of Cre activation having less starting Cre molecules can be bypassed by higher light intensities or longer exposure times. In support of this, the data in **Rebuttal Figure 2** (see below) confirmed how in C26 cells, longer exposure times to 405 nm (30 vs 90 seconds) also provided a significant better signal-to-noise ratio.

Rebuttal Figure 2. FACS quantification of DiLiCre1.0-mediated recombination in C26 cells stably expressing the memMTQ2-nRFP reporter and sorted for low, intermediate and high DiLiCre1.0 expression. The three sorted populations were tested for three different concentrations of doxycycline (0.01, 0.1, or 1 μM) and either non-exposed or exposed to light. Two different LED light regimes were tested: 0.5mW/mm² 30s-pulses every 4 hours [x3] (panel d), and 0.5mW/mm² 90s-pulses every 4 hours [x3] (panel e). Data represents the mean±SD from n=3 biologically independent experiments. Control values (untreated cells) were subtracted. Right graphs represent the dynamic range calculations (signal-to-noise ratios) for panels d and e, respectively.

Fig. 1d: it'd be better if the authors could provide quantification of nRFP+ cells to demonstrate the efficiency of DiLiCre 1.0.

Reply:

This information has been added to **Extended Data Fig. 3a** of the revised manuscript and text modified accordingly in page 5 lines 122.

Fig. 2d: it'd be better if the authors added the raw flowcytometry results as they did for Fig. 3b.

Reply:

This information has been added to **Extended Data Fig. 4** of the revised manuscript and text modified accordingly in page 7 lines 172.

Fig. 3b: the flowcytometry images are too small. It'd be better to move the whole images to supplementary material using larger images with better resolution.

Reply:

As requested, we have moved this data to the supplementary section (**Extended Data Fig. 6a** of the revised manuscript), and text modified accordingly in page 8 lines 189.

Fig. 5c,d It'd be better to analyze the number of GFP+ cells as well as GFP intensity because other area seem to have GFP+ cells.

Reply:

As requested, we have included this analysis in **Extended Data Fig. 9d** of the revised manuscript.

Reviewer #2:

Cre recombinase is a valuable tool for making genetic modifications in live animals. Several light activated Cre recombinases have been reported. These systems could potentially be used for induction of recombination events in specific tissues and at specific times, which could enable new studies of physiology, development, cancer, etc. However, previous light activated Cre systems exhibit substantial leakiness in the dark, which may have prevented their deployment in vivo.

The authors address this challenge by engineering a lentiviral expression system wherein a recent violet-light activated engineered Cre protein is expressed from a stringent doxycycline inducible promoter. They further increase Cre activity while reducing leakiness by altering the engineered domain architecture of the photactivatable Cre protein in several ways. The performance is improved relative to previous systems, though there is still leakiness, especially in vitro. However, the authors find that the system enables fairly strong optical activation with relatively low impact of leakiness in vivo settings including lineage tracing and oncogenesis. By inducing an oncogenic gene, they make some interesting observations of potential cell migratory processes in cancer.

Overall, I think this is an interesting and impactful study. Other groups should find this tool useful for in vivo studies. I have some concerns with the manuscript, which are detailed below. If this concerns can be addressed, I recommend publication in Nature Communications.

Reply:

We thank this reviewer for his/her time to review our manuscript and for his/her supporting words about our work. Below, we will address all concerns point-by-point.

Concerns

It is not clear to me how PhoCl photoactivation causes Cre recombinase to become dissociated from the ERT domains so that it can translocate to the nucleus. The authors should elaborate on the mechanism of this process.

Reply:

Thanks for pointing this out. The mechanism of the photocleave of PhoCl and the subsequent dissociation of the protein is well described by several key papers in the field (e.g. Mizuno et al., Mol cell 2003; Davidson and Campbel, Nat Meth 2009; Lu et al, Chem Sci 2021). When illuminated with violet light (~405 nm), the green chromophore of PhoCl undergoes a main chain-breaking β -elimination, leading to red signal. Subsequently, the fluorophore dissociates leading to loss of fluorescence. In the revised manuscript, we have included this description at lines 94-95 at page 5. Moreover, we have included a cartoon that contains this information (**Extended Data Fig. 1a-to-c** of the revised manuscript).

The authors claim that the green-to-red switch of PhoCl enables “optimization of light exposure, rendering it ideal for our experiments” (line, 92). I do not understand what the authors mean by this statement. What about light exposure needs to be optimized? What does optimal mean in this context? How does PhoCl enable this optimization, compared to other photoswitchable domains used to control Cre?

Reply:

To our knowledge, there are no other photoswitchable domains that can break into two pieces by light and therefore can be used to control Cre activity.

Tissues cause light to scatter, and therefore it is difficult to expose cells to a well-controlled amount of light. This is important, since too little light may not induce enough Cre activity and too much light may cause bleaching of PhoCL unit of DiLiCre2.0 without inducing photoswithcing/activating of Cre. So instead of taking an optimized activation light amount (which is not possible *in vivo*), we take a different approach and optimize light-exposure for each experiment by following the green-to-red switch of the PhoCl unit of DiLiCre2.0, and stop the exposure of cells to light when we do not longer observe a red-to-green switch. In the revised manuscript, we have clarified this point better at lines 94-95 at page 5.

The authors show that DiLiCre0.0 is photosilent, and go on to fix it by removing the N-terminal ERT2-PhoCl domain, which is nice. However, they provide no explanation for why this domain was contributing to photosilencing, nor why they chose to remove it as opposed to the C-terminal domain. The authors should at least present a hypothesis for why version 0.0 does not work and how removal of this domain restores function.

Reply:

We also extensively debated this topic in the lab. We came to the conclusion that at medium light intensities (i.e. intensities that do not cause phototoxicity), only a fraction of all ERT2-PhoCl units are cleaved so that in most DiLiCre0.0 only one of the ERT2-PhoCl domains is released (either the N or the C terminal) whilst the non-cleaved ERT2 domain keeps the DiLiCre inactive. If true, by removing one of the two PhoCl domains, regardless of whether it is the N or the C terminal one, should improve the photoactivation efficiency. The choice to remove the N-terminal domain was only based on technical reasons, since the cloning step to remove the PhoCl was much easier for the N than for the C terminus. Indeed, we found that removing one of the ERT2-PhoCl domains increased the efficiency to activate DiLiCre (see **Extended Data Figure 3e-to-j** the revised manuscript).

The timelapse data in Figure 1d is not clearly presented. There are numbers with increasing values overlaid on images of cells. However, those numbers have no units, and are not defined in the figure legend. The presentation should be clearer.

Reply:

Thanks for noticing this omission. Those numbers represent hours and minutes (hh:mm). In the revised manuscript, we have defined these units in the figure legend (**Figure 1d** of the revised manuscript).

Also, the response of the cells seems much stronger in panel c than in panel d. The authors state that panel d is a timelapse series that led to the data in panel c. Why is the response so much weaker in panel d? Any differences between these two experiments should be clarified.

Reply:

Upon photoconversion, the ERT2-PhoCl unit breaks off and the Cre unit in the DiLiCre gets activated (see cartoon in **Extended Data Fig 1**). However, the Cre mediated recombination is a relative inefficient process and therefore not every photoactivated DiLiCre molecule will lead to recombination. **Figure 1 panel c** shows the photoconversion/activation of the PhoCl unit of DiLiCre. **Figure 1 panel d** illustrates the subsequent recombination event.

I don't understand the statement on lines 126-128. Shouldn't removal of an ERT2 domain result in a decrease in cytoplasmic sequestration of Cre?

Reply:

Thanks for noticing this typo. It should say decrease in "cytoplasmic" not "nuclear" sequestering. We have corrected the text accordingly in the revised manuscript.

The authors do not define how the cytometry data are plotted in the figure legends (e.g. Extended Data Figure 2f). What are the multiple data points in each data set?

Reply:

In this figure, we analyzed Cre mediated recombination for different populations of HEK293T cells transduced with DiLiCre1.0 construct by flow cytometry. Three different populations of HEK293T cells were tested: expressing low, intermediate, and high levels of DiLiCre1.0. We tested each population separately in three experimental conditions: untreated cells, cells non exposed to light but treated with doxycycline (three different concentrations were addressed: 10, 100 or 1000 nM, blue data points), and cells co-treated with doxycycline and light (red data points). In the revised manuscript, we have added this description in the legend of **Extended Data Fig. 3j**.

The claim that removal of one of the two ERT2 domains from DiLiCre0.0 leads to a significant amount of background activity (lines 139-140) is not supported by the data provided (Extended Data Fig. 2f). The authors show no data comparing DiLiCre0.0 to DiLiCre1.0, they only compare different expression levels of the latter.

Reply:

Thanks for pointing this out. We have now included this data in in **Extended Data Figure 3i** of the revised manuscript.

Figure 3b is not readable. There is far too much information shown. The feature and text sizes are extremely small. The presentation needs to be reworked. All of the raw data that is currently being shown could be presented in the Extended Data.

Reply:

In the revised manuscript, we have moved all FACS data to **Extended Data Figure 6a**.

How is it that the fold change of recombination (Figure 2c) is so much higher in in vivo (~20-fold; Figure 3c) than in vitro (~2-fold; Figure 2d).

Reply:

In vitro we have systems where, even though we do lentiviral titrations and measured the levels of recombination at different conditions (e.g., different levels of expression of DiLiCre), we never have only one copy of DiLiCre per cell. Therefore, in *in vitro* experiments, only meant for the optimization of the paCre tools, we work with overexpression systems, which render higher levels of background recombination and therefore lower signal-to-noise ratios. Actually, by comparing the results of the reviewer's mentioned figures, one can observed that the recombination levels are quite comparable and that the only parameter that really changes is the background recombination. Higher background recombination levels have of course a clear impact when calculating the fold changes.

On the other hand, *in vivo*, we have one DiLiCre2.0 copy per cell (as our SouthernBlot analysis confirmed). This contributes to lower background recombination levels and therefore better dynamic range values.

I do not understand the treatment conditions in Figures 3d and e. For example, what do the 3 columns in Figure 3d represent? Different times of light exposure? If that's the case, then what is meant by "light control" in row 1? Does that actually mean no dox was added? The presentation of conditions in these figure panels should be clarified.

Reply:

For **Figure 3**, we have imaged organoids over time (i.e. taking images of the organoids at various time points). Hence, the three columns in **Figure 3c** represent different time point of these time lapses. We meant with "light control" the condition in which the organoids were exposed to light in the absence of doxycycline. In hence, this may not be the best terminology, and in the revised manuscript we now refer to "Light-exposed, - Dox". To further avoid confusion, we have now changed the text above the images, which now reads: 0 hours, 1 hour, etc.

Reviewer #3:

Regulation of gene expressions in a spatiotemporal manner is required to understand the role of the specific gene product and related biological mechanisms. However, most chemical inducers are toxic to animals and CRISPR-based systems are hard to deliver guide RNA to a target organ. To detour the limitations of existing genome inducer tools, the authors improved the lightinducible Cre recombinase system.

The authors introduced a doxycycline-inducible TETon promoter to the existing PhoCl system to create the two-lock photoactivatable Cre recombinase (DiLiCre1.0). Further fusion of the

additional ERT2 domain reduced the background recombination without a loss of efficiency (DiLiCre2.0). DiLiCre2.0 is activated by short light exposure times with an optimized ratio of light-induced and background Cre activity. Additionally, DiLiCre2.0 was characterized in cancer cells and organoid systems.

Finally, in vivo induction of mutagenesis and following cell-tracing is possible using DiLiCre2.0. Authors induced spatiotemporal HrasV12 mutagenesis in basal layer cells. They observed that the HrasV12 mutation had cells migrate toward the collagen dermal layer, which is the opposite behavior compared to normal cells of the basal layer. Such positional tracing of skin cells showed the applicability of DiLiCre2.0 to in vivo cell-tracing which enables spatiotemporal observation of biological phenomena.

Pros.

1) Introduction of doxycycline-inducible TETon system into developed photoactivatable Cre improves limitations of existing light-inducible Cre recombinases by reducing background activity.

2) Additional control of light-inducible Cre enables spatiotemporal regulation of Cre recombinase and following recombination in vivo in the skin and intestinal crypt.

3) Positional cell tracing of a basal and collagen layer with DiLiCre2.0 knock-in mice suggests the applicability of the light-induced Cre system for lineage tracing of mutated cells in the skin.

Cons.

1) It is hard to estimate the relative efficiency and background activity of DiLiCre2.0 due to the lack of comparison of DiLiCre2.0 with other developed photoactivatable Cre systems.

Reply:

We would like to thank this reviewer for his/her time to review our manuscript and for the supportive words about our work. Below we address each concern point-by point.

1. In Extended Data Fig.1j, pCMV-Cre induced a conversion from membranous mTq2 signal into nuclear turboRFP signal. There was an obvious decrease in mTq2 signal while DiLiCre1.0 failed to diminish the mTq2 signal after light-activation in Figure.1d. We recommend measuring the intensity change of membranous blue signal (pre-conversion versus post-conversion) to fully support that the RFP signal emerged by Cre recombinase and following recombination. There should be a mTq2 intensity difference even if such difference was generated by a difference of efficiency between pCMV-Cre and DiLiCre1.0.

Reply:

For the revised manuscript, we have measured the memMTQ2 intensity (**Extended Data Figure 3b and c** of the revised manuscript), comparing recombinant and non-recombinant cells, and our results confirm that once recombination is initiated, the memMTQ2 expression levels drop significantly in the photoexposed areas in cells where nRFP emerges. The plot contains data related to **Figure 1d** (HEK293T cells) from three independent positions and images are representative images of three different regions within one of those imaged positions at experimental endpoint (T65.28, hh:mm).

2. In Figure.2c and 2d, DiLiCre2.0 showed better performance than DiLiCre1.0 for Cre activity. However, I questioned the relative efficiency and background Cre activity compared to other existing photoactivatable Cre recombinases, such as paCre and PhoCl, as you mentioned. It would be more powerful to display the performance of DiLiCre2.0 with other developed photoactivatable Cre recombinase tools to argue that DiLiCre2.0 well-overcame presenting limitations.

Reply:

There are two different photoactivatable Cre (PA-Cre) systems: 1) based on dimerization (e.g. Magnet, split-Cre), 2) based on cleavage (PhoCl). Our DiLiCre system belongs to the second category. The dimerization series is activated by blue light (460-480nm), whilst the PhoCl series is activated by violet light (~405nm). Therefore, a direct comparison between those series using the same illumination and reporters is not possible. Instead, we investigated the various characteristics that are required for successful use of these PA-Cre series for mouse experiments.

Comparison to dimerization systems:

Although different activation wavelengths between dimerization based photoactivatable Cre systems and our DiLiCre2.0 system renders a side-by-side comparison difficult, for the revised manuscript we have compared some basic traits. We tested two different split-Cre systems: the Magnet system by the Sato group (Kawano et al, 2016) and the PA-cre2.0 by the Tucker group (Meador et al., 2019). Both are based on a split-Cre strategy, where the Cre subunits are each fused to blue-light sensitive dimerization domains that reconstitute Cre enzymatic activity. The Magnet system published by the Sato group (Kawano et al, 2016) is based on fungal dimerizing proteins, while the PA-Cre2.0 by the Tucker lab (Meador et al, 2019) is based on Arabidopsis proteins. Both were reported to have a signal to noise ratios >200 (defined as the ratio between Cre activity in the presence of 460-480nm blue light versus activity in the dark).

We first tested the Magnet system, as it was the first to be published and was reported to have excellent signal/noise figures. However, Duplus-Bottin and colleagues (2021) recently published that the Magnet system in their hands led to much less favorable ratio of photo-induced Cre-activity and background Cre activity than reported in the original report. Indeed, our transfection experiments suggested higher levels of background recombination than those reported by the original manuscript (**Extended data Fig. 12a** of the revised manuscript). We then speculated that constitutive expression could be responsible for the elevated background noise, so we tested whether an inducible construct would solve this problem (**Extended data Fig. 12b** of the revised manuscript). In our setting however, the overall recombination efficiency was far lower than the published results. Upon doxycycline treatment (100nM), background activation in the absence of light was unacceptably high, up to 35% of the total signal (**Extended data Fig. 12b** of the revised manuscript). We established that activation time was not the issue, as blue light irradiation times between 2 min and 20min minimally affected Magnet efficiency (**Extended data Fig. 12b** of the revised manuscript). Therefore, our observations are in line with other recent reports (Duplus-Bottin et al., 2021) that the signal-to-noise ratio of the Magnet is too low for a practical tool.

We then tested the PA-Cre2.0 system published by the Tucker lab in 2019 and followed a similar strategy. A side-by-side comparison with the Magnet system showed that this construct had an intrinsic lower noise (**Extended data Fig. 12a** versus **c** of the revised manuscript). Although *in vitro* PA-Cre2.0 Tucker had better overall efficiency and reduced noise compared to the Magnet system, this system showed some ground problems preventing *in vitro* and *in vivo* use: we were unable to stably integrate or avoid the silencing of the PA-Cre2.0 in HEK293T cells as it was found by our multiple rounds of FACS enrichment for PA-Cre2.0 (**Extended data Fig. 12d** of the revised manuscript). More than 90% of the mCherry-PA-Cre2.0 positive cells selected in first round were lost upon few cell passages. We reported this observation to Tucker's lab and they confirmed that they experienced the same problems and their unsuccessful attempts to get stable integrations.

From those experiments, we concluded that the published dimerization-based photoactivatable Cre are not usable for *in vivo* use because 1) too much background, 2) too low recombination efficiency, 3) cannot generate stable lines (due to the lack of permanent integration or silencing).

Lastly, and not unimportantly, the blue activation light possess a large limitation for the *in vivo* use of dimerization systems. Most reporter models that are currently available are based on GFP-like fluorescence that require blue excitation light. This blue excitation light also activated the dimerization systems rendering it unusable for Cre lineage tracing experiments.

In the revised manuscript, we describe the limitations of these dimerization systems at page 10 and lines 265-269.

Comparison to photocleavable (PhoCI) based systems:

We also compared our DiLiCre2.0 with the PhoCI series. We have tested three modified versions of the original photocleavable Cre system containing the 405nm-photoswitchable protein PhoCI (Zhang et al., 2017). Similar to our strategy when testing the dimerizing systems (Magnet and PA-Cre2.0), all our adapted versions of the photocleavable Cre system worked under TetOn response elements and doxycycline induction in order to minimize the background recombination levels.

First, our data shown that DiLiCre0.0, a doxycycline inducible version of the original construct used by Zhang and colleagues 2017, failed to induce efficient recombination of cells both in the absence of presence of light (**Extended Data Figure 2a-i** of the revised manuscript). However, DiLiCre1.0, which contains only one PhoCI-ERT2 tandem repeat, shown higher background recombination compared to DiLiCre0.0 (**Extended Data Figure 3a, e** and **i** of the revised manuscript) and most importantly, light-induced Cre recombinase activity (**Figure 1d** and **Extended Data Figure 3a, g** and **j** of the revised manuscript). The induction of DiLiCre1.0 with doxycycline at 100nM in HEK293T cells, revealed $8.04 \pm 1.99\%$ of background recombination and $25.44 \pm 3.60\%$ of light-induced Cre recombinase activity (**Extended Data Figure 3j** of the revised manuscript). In our attempt to improve the ratio of light-induced and background activity of the Cre recombinase, we developed DiLiCre2.0, which contains two ERT tandem repeats. This modification led to an increase in induction efficiency and a lower background activity compared to DiLiCre1.0 (**Figure 2a-to-d** of the revised manuscript).

Secondly, we were able to achieve stable integrations of all our DiLiCre developed constructs *in vitro* and ultimately DiLiCre2.0 *in vivo*. This represents an important advantage respect the PA-Cre Tucker construct series based on Arabidopsis light responding proteins.

Conclusion:

From all these experimental comparisons, we concluded that DiLiCre2.0 was the molecular tool of choice for lineage tracing experiments and we therefore have developed the new photoactivatable Cre transgenic mouse model. Indeed, our conclusion is confirmed by our successful *in vivo* experiments with this mouse model (**Figures 4-to-6**).

3. In Figure.4, the authors have emphasized the spatiotemporal control of DiLiCre2.0 activity in vivo. However, spatial-specific Cre activation is not mentioned in the article. Though Figure.4 legend indicates the spatial control with the word 'ROI', it would be better to point out the spatial-specific photoactivation in the main paragraph.

Reply:

Thanks, this is a good point. In the revised manuscript, we point this out at page 8 and lines 204-205.

4. In Figure.5c, the authors have selected four photoactivated regions and four control regions but there are five samples in the graph in Figure.5d. Also, the authors have mentioned in the Method that the GFP signal was measured in 4-5 ROIs. Is missing ROI in the other window? Additionally, I questioned whether delicate spatiotemporal control is possible with DiLiCre2.0 in vivo by activating the region 1-4 sequentially and inducing the recombination in different time window.

Reply:

Sorry for this confusion. This experiment is based on 3 mice: In the first and second mouse, 4 different regions have been photo-converted/activated, whilst in the third mouse 5 regions have been photo-converted/activated. Moreover, the same number of non-illuminated control regions were picked per mouse. To better explain this to the reader, we have adapted the text/legend at lines 230 and page 9 of the revised manuscript.

In a following up study, we are testing whether DiLiCre2.0 can be used in a delicate spatiotemporal manner. This is a whole project by itself and we feel that this is beyond the scope of the current study. Therefore, we changed the text indicating "spatial control" wherever we have previously mentioned "spatiotemporal".

5. How can authors distinguish green signals between DiLiCre2.0 and HrasV12 other than time point in figure.5? Does a line 228 ("the amount of fluorescence of HraV12 is on order of magnitude larger than the fluorescence of DiLiCre2.0") explain the distinguishable property of DiLiCre2.0 and HrasV12 signals?

Reply:

Yes, the difference in intensity is so large that it is easy to distinguish. Moreover, because we work with an inducible system, the green signal coming from DiLiCre2.0 vanished in the following 24-36 hours

after treatment which contributes too to a better read-out of the HrasV12-eGFP fluorescence. Moreover, the DiLiCre2.0 fluorescence is cytoplasmic while the HrasV12-eGFP fluorescence is present at all cell compartment, which also contributes to distinguish both green signals.

6. Was there any experimental try to express and activate the DiLiCre2.0 other than skin? I questioned any application possibility of DiLiCre2.0 in other organs deeper than skin and intestinal crypts.

Reply:

We extensively tested the expression of DiLiCre2.0 other than the skin and intestine. We observed good expression in the kidney, liver, breast, ovary, testis, and lung, in addition to the skin and intestinal tissues (**Extended Data Fig. 5** of the revised manuscript). To illustrate the power to use DiLiCre2.0 for lineage tracing experiments *in vivo*, we performed lineage tracing experiments in the skin and intestine by optimizing the expression level of DiLiCre2.0 (i.e. amount of doxycycline), and light exposure. If we want to perform experiments in one of the other tissues, this optimization steps needs to be performed again which, to our feeling, falls beyond the scope of the current manuscript. We acknowledge these points at lines 286-288 at page 11 of the revised manuscript.

Reviewers' Comments:

Reviewer #1:

Remarks to the Author:

This reviewer appreciated the authors' responsibility to the concerns and questions, addressing experimentally. However, a remaining major concern is insufficient benchmark experiment to compare existing photo/light-activated Cre recombinase using the identical experimental setup at the same time with the authors' new one. The new results shown in Fig. S12 misses the author's crypts. tool DiLiCre2.0 using TRE3GV and Cre reporter plasmids. In addition, the PA-Cre constructs tested are relatively outdated. Some updated versions are available even in the PA-Cre series (ex. Morikawa et al. Nat. Commun. 2020). Also, the other new ones described below seem to be promising and should be compared on the benchmark experiment to conclude that DiLiCre2.0 could work the best among the available photo/light-controllable Cre.

ex.

<https://elifesciences.org/articles/61268>

<https://www.nature.com/articles/s41467-020-17530-9>

<https://www.nature.com/articles/s41592-020-0774-3>

The outcomes from this proposed benchmark experiment will be so valuable for future users. It'd be also great if the author could move the benchmark experimental results to the main figure.

Reviewer #2:

Remarks to the Author:

The authors have addressed my comments in the revised manuscript. I believe these changes are sufficient.

Reviewer #3:

Remarks to the Author:

In the revised manuscript, the authors sufficiently improved the previous concerns.

Most questions were well explained with acceptable experimental results.

There is one minor concern to improve your manuscript.

1) It'd be better to add quantification data in Extended data Fig.12 to visualize the imaging results.

I expect that DiLiCre2.0 could be used in various scientific applications and recommend the publication in Nature Communications.

Reviewer #1:

This reviewer appreciated the authors' responsibility to the concerns and questions, addressing experimentally.

Reply:

We thank this referee for reviewing our revised manuscript.

However, a remaining major concern is insufficient benchmark experiment to compare existing photo/light-activated Cre recombinase using the identical experimental setup at the same time with the authors' new one. The new results shown in Fig. S12 misses the author's tool DiLiCre2.0 using TRE3GV and Cre reporter plasmids. In addition, the PA-Cre constructs tested are relatively outdated. Some updated versions are available even in the PA-Cre series (ex. Morikawa et al. Nat. Commun. 2020). Also, the other new ones described below seem to be promising and should be compared on the benchmark experiment to conclude that DiLiCre2.0 could work the best among the available photo/light-controllable Cre.

ex.

<https://elifesciences.org/articles/61268>

<https://www.nature.com/articles/s41467-020-17530-9>

<https://www.nature.com/articles/s41592-020-0774-3>

The outcomes from this proposed benchmark experiment will be so valuable for future users. It'd be also great if the author could move the benchmark experimental results to the main figure.

Reply:

We appreciate the referee's request to extend our *in vitro* benchmark experiments of DiLiCre to three additional Cre recombinases. At first sight this sounds fair, but as we will discuss below, these requested additional experiments are out of the scope of our manuscript and will not add valuable information to our study.

Our aim is to develop a mouse model with a photoactivatable Cre that can be used for *in vivo* lineage tracing experiments. This requires particular characteristics (see below) that existing photo-activatable Cre models (including the ones that Reviewer #1 mentions) do not possess. Indeed, so far, no mouse models based on photo-activatable Cre have been published that can be used for lineage tracing experiments with meaningful biological insights.

Supported by our benchmark experiments (and well appreciated by Referees #2 and #3), our revised manuscript illustrates that the characteristics of DiLiCre are optimal for developing a mouse model for *in vivo* experiments: 1) the system does not cause a (lethal) phenotype in mice, 2) the background activity is minimized since DiLiCre is only expressed upon doxycycline, 3) for each organ/tissue of interest, the ratio of activation and background activity can be optimized by tuning the expression level of DiLiCre using various concentration of doxycycline, 4) the optimal exposure to the activating light (both in intensity and time) can be optimized by the green to red conversion of DiLiCre. This is particularly important *in vivo* since light scattering in tissues renders light intensity in the focal plane unknown. Lastly, 5) the green-to-red switch illustrates which cells have been exposed to light, which is critical for *in vivo* experiments where tissue movements during imaging (and thus during activation) are common.

The reviewer is right that there are different existing photo/light-activatable Cre models with different characteristics. However, none of those models, including the ones that the reviewer suggests, possesses all the above-mentioned characteristics required for *in vivo* experiments. The question is whether an *in vitro* comparison of characteristics unrelated to traits required for *in vivo* experiments, would make our manuscript stronger. An extensive *in vitro* comparison between different systems, as previously done (Kawano et al., 2016; Meador et al., 2019; Morikawa et al., 2020; Duplus-Bottin et al., 2022), are indeed nice resources for the field to understand which model to use for different *in vitro* applications. However, this does not provide any relevant information about their *in vivo* performance. Obviously, this can only be done by benchmark experiments in mice. However, this requires the generation of a mouse model for each existing photo-activatable Cre to compare their performance *in vivo*, which is out of the scope of our manuscript.

To better explained the unique characteristics of DiLiCre that renders it superior to other published systems for a mouse model and *in vivo* experiments, we adapted the discussion of the revised manuscript. At lines 261 to 270, we explain the unique characteristics that DiLiCre2.0 has that renders this system optimal for *in vivo* use. At line 282 to 284 we state that for comparing *in vivo* performance, benchmark experiments of different photoactivatable Cre systems need to be performed in mice, which requires the future efforts to generate mouse models of other systems.

Reviewer #2:

The authors have addressed my comments in the revised manuscript. I believe these changes are sufficient.

Reply:

We thank this referee for his/her supporting words to publish our manuscript in Nature Communications.

Reviewer #3:

In the revised manuscript, the authors sufficiently improved the previous concerns. Most questions were well explained with acceptable experimental results. "I expect that DiLiCre2.0 could be used in various scientific applications and recommend the publication in Nature Communications."

Reply:

We also thank this referee for his/her support to publish our work in Nature Communications.

There is one minor concern to improve your manuscript. 1) It'd be better to add quantification data in Extended data Fig.12 to visualize the imaging results.

Reply:

We have added this quantification in Extended data Fig 12 of the revised manuscript.